# Discovering Diverse Multi-Agent Strategic Behavior via Reward Randomization

**Zhenggang Tang**[*16†], **Chao Yu**[*1♯], **Boyuan Chen**[3], **Huazhe Xu**[3], **Xiaolong Wang**[4],
**Fei Fang**[5], **Simon Du**[7], **Yu Wang**[1], **Yi Wu**[12♯]

[1] Tsinghua University, [2] Shanghai Qi Zhi Institute, [3] UC Berkeley, [4] UCSD, [5] CMU,
[6] Peking University, [7] University of Washington

♯zoeyuchao@gmail.com, ♮jxwuyi@gmail.com

## Abstract

We propose a simple, general and effective technique, *Reward Randomization* for discovering diverse strategic policies in complex multi-agent games. Combining reward randomization and policy gradient, we derive a new algorithm, *Reward-Randomized Policy Gradient (RPG)*. RPG is able to discover multiple distinctive human-interpretable strategies in challenging temporal trust dilemmas, including grid-world games and a real-world game *Agar.io*, where multiple equilibria exist but standard multi-agent policy gradient algorithms always converge to a fixed one with a sub-optimal payoff for every player even using state-of-the-art exploration techniques. Furthermore, with the set of diverse strategies from RPG, we can (1) achieve higher payoffs by fine-tuning the best policy from the set; and (2) obtain an adaptive agent by using this set of strategies as its training opponents. The source code and example videos can be found in our website: https://sites.google.com/view/staghuntrpg.

## 1 Introduction

Games have been a long-standing benchmark for artificial intelligence, which prompts persistent technical advances towards our ultimate goal of building intelligent agents like humans, from Shannon's initial interest in Chess (Shannon, 1950) and IBM DeepBlue (Campbell et al., 2002), to the most recent deep reinforcement learning breakthroughs in Go (Silver et al., 2017), Dota II (OpenAI et al., 2019) and Starcraft (Vinyals et al., 2019). Hence, analyzing and understanding the challenges in various games also become critical for developing new learning algorithms for even harder challenges.

Most recent successes in games are based on decentralized multi-agent learning (Brown, 1951; Singh et al., 2000; Lowe et al., 2017; Silver et al., 2018), where agents compete against each other and optimize their own rewards to gradually improve their strategies. In this framework, Nash Equilibrium (NE) (Nash, 1951), where no player could benefit from altering its strategy unilaterally, provides a general solution concept and serves as a goal for policy learning and has attracted increasingly significant interests from AI researchers (Heinrich & Silver, 2016; Lanctot et al., 2017; Foerster et al., 2018; Kamra et al., 2019; Han & Hu, 2019; Bai & Jin, 2020; Perolat et al., 2020): many existing works studied how to design practical multi-agent reinforcement learning (MARL) algorithms that can provably converge to an NE in Markov games, particularly in the zero-sum setting.

Despite the empirical success of these algorithms, a fundamental question remains largely unstudied in the field: even if an MARL algorithm converges to an NE, ***which equilibrium will it converge to?*** The existence of multiple NEs is extremely common in many multi-agent games. Discovering as many NE strategies as possible is particularly important in practice not only because different NEs can produce drastically different payoffs but also because when facing unknown players who are trained to play an NE strategy, we can gain advantage by identifying which NE strategy the opponent is playing and choosing the most appropriate response. Unfortunately, in many games where multiple distinct NEs exist, the popular decentralized policy gradient algorithm (PG), which has led to great successes in numerous games including Dota II and Stacraft, always converge to a particular NE with non-optimal payoffs and fail to explore more diverse modes in the strategy space.

Consider an extremely simple example, a 2-by-2 matrix game *Stag-Hunt* (Rousseau, 1984; Skyrms, 2004), where two pure strategy NEs exist: a "risky" cooperative equilibrium with the highest payoff

---

*Equal contribution. † Work done as an intern at Institute for Interdisciplinary Information Sciences (IIIS), Tsinghua University.

for both agents and a "safe" non-cooperative equilibrium with strictly lower payoffs. We show, from both theoretical and practical perspectives, that even in this simple matrix-form game, PG fails to discover the high-payoff "risky" NE with high probability. The intuition is that the neighborhood that makes policies converge to the "risky" NE can be substantially small comparing to the entire policy space. Therefore, an exponentially large number of exploration steps are needed to ensure PG discovers the desired mode. We propose a simple technique, *Reward Randomization* (RR), which can help PG discover the "risky" cooperation strategy in the stag-hunt game with theoretical guarantees. The core idea of RR is to directly perturb the reward structure of the multi-agent game of interest, which is typically low-dimensional. RR directly alters the landscape of different strategy modes in the policy space and therefore makes it possible to easily discover novel behavior in the perturbed game

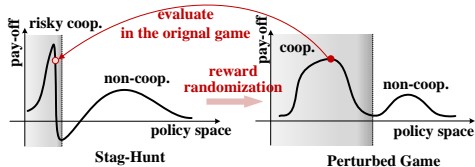

Figure 1: Intuition of Reward Randomization

(Fig. 1). We call this new PG variant *Reward-Randomized Policy Gradient* (RPG).

To further illustrate the effectiveness of RPG, we introduce three Markov games – two gridworld games and a real-world online game *Agar.io*. All these games have multiple NEs including both "risky" cooperation strategies and "safe" non-cooperative strategies. We empirically show that even with state-of-the-art exploration techniques, PG fails to discover the "risky" cooperation strategies. In contrast, RPG discovers a surprisingly *diverse* set of human-interpretable strategies in all these games, including some non-trivial emergent behavior. Importantly, among this set are policies achieving much higher payoffs for each player compared to those found by PG. This "diversity-seeking" property of RPG also makes it feasible to build *adaptive policies*: by re-training an RL agent against the diverse opponents discovered by RPG, the agent is able to dynamically alter its strategy between different modes, e.g., either cooperate or compete, w.r.t. its test-time opponent's behavior.

We summarize our contributions as follow

- We studied *a collection of challenging multi-agent games*, where the popular multi-agent PG algorithm always converges to a sub-optimal equilibrium strategy with low payoffs.

- *A novel reward-space exploration technique*, reward randomization (RR), for discovering hard-to-find equilibrium with high payoffs. Both theoretical and empirical results show that reward randomization substantially outperforms classical policy/action-space exploration techniques in challenging trust dilemmas.

- We empirically show that RR *discovers surprisingly diverse strategic behaviors in complex Markov games*, which further provides a practical solution for building an adaptive agent.

- *A new multi-agent environment Agar.io*, which allows complex multi-agent strategic behavior. We released the environment to the community as a novel testbed for MARL research.

## 2  A MOTIVATING EXAMPLE: STAG HUNT

We start by analyzing a simple problem: finding the NE with the optimal payoffs in the *Stag Hunt* game. This game was originally introduced in Rousseau's work, "*A discourse on inequality*" (Rousseau, 1984): a group of hunters are tracking a big stag silently; now a hare shows up, each hunter should decide whether to keep tracking the stag or kill the hare immediately. This leads to the 2-by-2 matrix-form stag-hunt game in Tab. 1

|  | Stag | Hare |
|---|---|---|
| Stag | $a, a$ | $c, b$ |
| Hare | $b, c$ | $d, d$ |

Table 1: The stag-hunt game, $a > b \geq d > c$.

with two actions for each agent, *Stag* (S) and *Hare* (H). There are two pure strategy NEs: the Stag NE, where both agents choose S and receive a high payoff $a$ (e.g., $a = 4$), and the Hare NE, where both agents choose H and receive a lower payoff $d$ (e.g., $d = 1$). The Stag NE is "risky" because if one agent defects, they still receives a decent reward $b$ (e.g., $b = 3$) for eating the hare alone while the other agent with an S action may suffer from a big loss $c$ for being hungry (e.g., $c = -10$).

Formally, let $A = \{S, H\}$ denote the action space, $\pi_i(\theta_i)$ denote the policy for agent $i$ ($i \in \{1, 2\}$) parameterized by $\theta_i$, i.e., $P[\pi_i(\theta_i) = S] = \theta_i$ and $P[\pi_i(\theta_i) = H] = 1 - \theta_i$, and $R(a_1, a_2; i)$ denote the payoff for agent $i$ when agent 1 takes action $a_1$ and agent 2 takes action $a_2$. Each agent $i$ optimizes its expected utility $U_i(\pi_1, \pi_2) = \mathbb{E}_{a_1 \sim \pi_1, a_2 \sim \pi_2}[R(a_1, a_2; i)]$. Using the standard policy gradient algorithm, a typical learning procedure is to repeatedly take the following two steps until

convergence[1]: (1) estimate gradient $\nabla_i = \nabla U_i(\pi_1, \pi_2)$ via self-play; (2) update the policies by $\theta_i \leftarrow \theta_i + \alpha \nabla_i$ with learning rate $\alpha$. Although PG is widely used in practice, the following theorem shows in certain scenarios, unfortunately, the probability that PG converges to the Stag NE is low.

**Theorem 1.** *Suppose $a - b = \epsilon(d - c)$ for some $0 < \epsilon < 1$ and initialize $\theta_1, \theta_2 \sim \text{Unif}[0,1]$. Then the probability that PG discovers the high-payoff NE is upper bounded by $\frac{2\epsilon + \epsilon^2}{1 + 2\epsilon + \epsilon^2}$.*

Theorem 1 shows when the risk is high (i.e., $c$ is low), then the probability of finding the Stag NE via PG is very low. Note this theorem applies to random initialization, which is standard in RL.

**Remark:** *One needs at least $N = \Omega\left(\frac{1}{\epsilon}\right)$ restarts to ensure a constant success probability.*

Fig. 2 shows empirical studies: we select 4 value assignments, i.e., $c \in \{-5, -20, -50, -100\}$ and $a=4$, $b=3$, $d=1$, and run a state-of-the-art PG method, proximal policy optimization (PPO) (Schulman et al., 2017), on these games. The Stag NE is rarely reached, and, as $c$ becomes smaller, the probability of finding the Stag NE significantly decreases. Peysakhovich & Lerer (2018b) provided a theorem of similar flavor without analyzing the dynamics of the learning algorithm whereas we explicitly characterize the behavior of PG. They studied a prosocial reward-sharing scheme, which transforms the reward of both agents to $R(a_1, a_2; 1) + R(a_1, a_2; 2)$. Reward sharing can be viewed as a special case of our method and, as shown in Sec. 5, it is insufficient for solving complex temporal games.

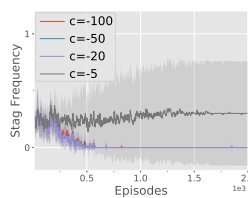

Figure 2: PPO in stag hunt, with $a=4$, $b=3$, $d=1$ and various $c$ (10 seeds).

## 2.1 Reward Randomization in the Matrix-Form Stag-Hunt Game

9 Thm. 1 suggests that the utility function $R$ highly influences what strategy PG might learn. Taking one step further, even if a strategy is difficult to learn with a particular $R$, it might be easier in some other function $R'$. Hence, if we can define an appropriate space $\mathcal{R}$ over different utility functions and draw samples from $\mathcal{R}$, we may possibly discover desired novel strategies by running PG on some sampled utility function $R'$ and evaluating the obtained policy profile on the original game with $R$. We call this procedure *Reward Randomization* (RR).

Concretely, in the stag-hunt game, $R$ is parameterized by 4 variables $(a_R, b_R, c_R, d_R)$. We can define a distribution over $\mathbb{R}^4$, draw a tuple $R' = (a_{R'}, b_{R'}, c_{R'}, d_{R'})$ from this distribution, and run PG on $R'$. Denote the original stag-hunt game where the Stag NE is hard to discover as $R_0$. Reward randomization draws $N$ perturbed tuples $R_1, \ldots, R_N$, runs PG on each $R_i$, and evaluates each of the obtained strategies on $R_0$. The theorem below shows it is highly likely that the population of the $N$ policy profiles obtained from the perturbed games contains the Stag NE strategy.

**Theorem 2.** *For **any** Stag-Hunt game, suppose in the $i$-th run of RR we randomly generate $a_{R_i}, b_{R_i}, c_{R_i}, d_{R_i} \sim \text{Unif}[-1,1]$ and initialize $\theta_1, \theta_2 \sim \text{Unif}[0,1]$, then with probability at least $1 - 0.6^N = 1 - \exp(-\Omega(N))$, the aforementioned RR procedure discovers the high-payoff NE.*

Here we use the uniform distribution as an example. Other distributions may also help in practice. Comparing Thm. 2 and Thm. 1, RR significantly improves standard PG w.r.t. success probability.

**Remark 1:** *For the scenario studied in Thm. 1, to achieve a $(1 - \delta)$ success probability for some $0 < \delta < 1$, PG requires **at least** $N = \Omega\left(\frac{1}{\epsilon} \log\left(\frac{1}{\delta}\right)\right)$ random restarts. For the same scenario, RR only requires to repeat **at most** $N = O\left(\log(1/\delta)\right)$ which is independent of $\epsilon$. When $\epsilon$ is small, this is a huge improvement.*

**Remark 2:** *Thm. 2 suggests that comparing with policy randomization, perturbing the payoff matrix makes it substantially easier to discover a strategy that can be hardly reached in the original game.*

Note that although in Stag Hunt, we particularly focus on the Stag NE that has the highest payoff for both agents, in general RR can also be applied to NE selection in other matrix-form games using a payoff evaluation function $E(\pi_1, \pi_2)$. For example, we can set $E(\pi_1, \pi_2) = U_1(\pi_1, \pi_2) + U_2(\pi_1, \pi_2)$ for a prosocial NE, or look for Pareto-optimal NEs by setting $E(\pi_1, \pi_2) = \beta U_1(\pi_1, \pi_2) + (1 - \beta)U_2(\pi_1, \pi_2)$ with $0 \le \beta \le 1$.

---

[1]In general matrix games beyond stag hunt, the procedure can be cyclic as well (Singh et al., 2000).

---
**Algorithm 1:** RPG: Reward-Randomized Policy Gradient

---
**Input:** original game $M$, search space $\mathcal{R}$, evaluation function $E$, population size $N$;

draw samples $\{R^{(1)}, \ldots, R^{(N)}\}$ from $\mathcal{R}$;

$\{\pi_1^{(i)}, \pi_2^{(i)}\} \leftarrow$ PG on induced games $\{M(R^{(i)})\}_i$ in parallel ;            // RR phase

select the best candidate $\pi_1^{(k)}, \pi_2^{(k)}$ by $k = \arg\max_i E(\pi_1^{(i)}, \pi_2^{(i)})$ ; // evaluation phase

$\pi_1^{\star}, \pi_2^{\star} \leftarrow$ fine-tune $\pi_1^{(k)}, \pi_2^{(k)}$ on $M$ via PG (if necessary) ;       // fine-tuning phase

**return** $\pi_1^{\star}, \pi_2^{\star}$;

---

## 3  RPG: Reward-Randomized Policy Gradient

Herein, we extend *Reward Randomization* to general multi-agent Markov games. We now utilize RL terminologies and consider the 2-player setting for simplicity. Extension to more agents is straightforward (Appx. B.3).

Consider a 2-agent Markov game $M$ defined by $(\mathcal{S}, \mathcal{O}, \mathcal{A}, R, P)$, where $\mathcal{S}$ is the state space; $\mathcal{O} = \{o_i : s \in \mathcal{S}, o_i = O(s, i), i \in \{1, 2\}\}$ is the observation space, where agent $i$ receives its own observation $o_i = O(s; i)$ (in the fully observable setting, $O(s, i) = s$); $\mathcal{A}$ is the action space for each agent; $R(s, a_1, a_2; i)$ is the reward function for agent $i$; and $P(s'|s, a_1, a_2)$ is transition probability from state $s$ to state $s'$ when agent $i$ takes action $a_i$. Each agent has a policy $\pi_i(o_i; \theta_i)$ which produces a (stochastic) action and is parameterized by $\theta_i$. In the decentralized RL framework, each agent $i$ optimizes its expected accumulative reward $U_i(\theta_i) = \mathbb{E}_{a_1 \sim \pi_1, a_2 \sim \pi_2}[\sum_t \gamma^t R(s^t, a_1^t, a_2^t; i)]$ with some discounted factor $\gamma$.

Consider we run decentralized RL on a particular a Markov game $M$ and the derived policy profile is $(\pi_1(\theta_1), \pi_2(\theta_2))$. The desired result is that the expected reward $U_i(\theta_i)$ for each agent $i$ is maximized. We formally written this equilibrium evaluation objective as an evaluation function $E(\pi_1, \pi_2)$ and therefore the goal is to find the optimal policy profile $(\pi_1^{\star}, \pi_2^{\star})$ w.r.t. $E$. Particularly for the games we considered in this paper, since every (approximate) equilibrium we ever discovered has a symmetric payoff, we focus on the empirical performance while assume a much simplified equilibrium selection problem here: it is equivalent to define $E(\pi_1, \pi_2)$ by $E(\pi_1, \pi_2) = \beta U_1(\theta_1) + (1 - \beta)U_2(\theta_2)$ for any $0 \le \beta \le 1$. Further discussions on the general equilibrium selection problem can be found in Sec. 6.

The challenge is that although running decentralized PG is a popular learning approach for complex Markov games, the derived policy profile $(\pi_1, \pi_2)$ is often sub-optimal, i.e., there exists $(\pi_1^{\star}, \pi_2^{\star})$ such that $E(\pi_1^{\star}, \pi_2^{\star}) > E(\pi_1, \pi_2)$. It will be shown in Sec. 5 that even using state-of-the-art exploration techniques, the optimal policies $(\pi_1^{\star}, \pi_2^{\star})$ can be hardly achieved.

Following the insights from Sec. 2, reward randomization can be applied to a Markov game $M$ similarly: if the reward function in $M$ poses difficulties for PG to discover some particular strategy, it might be easier to reach this desired strategy with a perturbed reward function. Hence, we can then define a reward function space $\mathcal{R}$, train a population of policy profiles in parallel with sampled reward functions from $\mathcal{R}$ and select the desired strategy by evaluating the obtained policy profiles in the original game $M$. Formally, instead of purely learning in the original game $M = (\mathcal{S}, \mathcal{O}, \mathcal{A}, R, P)$, we define a proper *subspace* $\mathcal{R}$ over possible reward functions $R : \mathcal{S} \times \mathcal{A} \times \mathcal{A} \to \mathbb{R}$ and use $M(R') = (\mathcal{S}, \mathcal{O}, \mathcal{A}, R', P)$ to denote the induced Markov game by replacing the original reward function $R$ with another $R' \in \mathcal{R}$. To apply reward randomization, we draw $N$ samples $R^{(1)}, \ldots, R^{(N)}$ from $\mathcal{R}$, run PG to learn $(\pi_1^{(i)}, \pi_2^{(i)})$ on each induced game $M(R^{(i)})$, and pick the desired policy profile $(\pi_1^{(k)}, \pi_2^{(k)})$ by calculating $E$ in the original game $M$. Lastly, we can fine-tune the policies $\pi_1^{(k)}, \pi_2^{(k)}$ in $M$ to further boost the practical performance (see discussion below). We call this learning procedure, *Reward-Randomized Policy Gradient* (RPG), which is summarized in Algo. 1.

**Reward-function space:**  In general, the possible space for a *valid* reward function is intractably huge. However, in practice, almost all the games designed by human have low-dimensional reward structures based on objects or events, so that we can (almost) always formulate the reward function in a linear form $R(s, a_1, a_2; i) = \phi(s, a_1, a_2; i)^T w$ where $\phi(s, a_1, a_2; i)$ is a low-dimensional feature vector and $w$ is some weight.

A simple and general design principle for $\mathcal{R}$ is to *fix the feature vector $\phi$ while only randomize the weight $w$*, i.e., $\mathcal{R} = \{R_w : R_w(s, a_1, a_2; i) = \phi(s, a_1, a_2; i)^T w, \|w\|_\infty \le C_{\max}\}$. Hence, the overall search space remains a similar structure as the original game $M$ but contains a diverse range of preferences over different feature dimensions. Notably, since the optimal strategy is invariant to the scale of the reward function $R$, theoretically any $C_{\max} > 0$ results in the same search space.

However, in practice, the scale of reward may significantly influence MARL training stability, so we typically ensure the chosen $C_{\max}$ to be compatible with the PG algorithm in use.

Note that a feature-based reward function is a standard assumption in the literature of inverse RL (Ng et al., 2000; Ziebart et al., 2008; Hadfield-Menell et al., 2017). In addition, such a reward structure is also common in many popular RL application domains. For example, in navigation games (Mirowski et al., 2016; Lowe et al., 2017; Wu et al., 2018), the reward is typically set to the negative distance from the target location $L_T$ to the agent's location $L_A$ plus a success bonus, so the feature vector $\phi(s, a)$ can be written as a 2-dimensional vector $[\|L_T - L_A\|_2, \mathbb{I}(L_T = L_A)]$; in real-time strategy games (Wu & Tian, 2016; Vinyals et al., 2017; OpenAI et al., 2019), $\phi$ is typically related to the bonus points for destroying each type of units; in robotics manipulation (Levine et al., 2016; Li et al., 2020; Yu et al., 2019), $\phi$ is often about the distance between the robot/object and its target position; in general multi-agent games (Lowe et al., 2017; Leibo et al., 2017; Baker et al., 2020), $\phi$ could contain each agent's individual reward as well as the joint reward over each team, which also enables the representation of different prosociality levels for the agents by varying the weight $w$.

**Fine tuning:** There are two benefits: (1) the policies found in the perturbed game may not remain an equilibrium in the original game, so fine-tuning ensures convergence; (2) in practice, fine-tuning could further help escape a suboptimal mode via the noise in PG (Ge et al., 2015; Kleinberg et al., 2018). We remark that a practical issue for fine-tuning is that when the PG algorithm adopts the actor-critic framework (e.g., PPO), we need an additional *critic warm-start phase*, which only trains the value function while keeps the policy unchanged, before the *fine-tuning phase* starts. This warm-start phase significantly stabilizes policy learning by ensuring the value function is fully functional for variance reduction w.r.t. the reward function $R$ in the original game $M$ when estimating policy gradients.

### 3.1 Learning to Adapt with Diverse Opponents

In addition to the final policies $\pi_1^\star, \pi_2^\star$, another benefit from RPG is that the population of $N$ policy profiles contains diverse strategies (more in Sec. 5). With a diverse set of strategies, we can build an adaptive agent by training with a random opponent policy sampled from the set per episode, so that the agent is forced to behave differently based on its opponent's behavior. For simplicity, we consider learning an adaptive policy $\pi_1^a(\theta^a)$ for agent 1. The procedure

---

**Algorithm 2:** Learning to Adapt

**Input:** game $M$, policy set $\Pi_2$, initial $\pi_1^a$;
**repeat**
    draw a policy $\pi_2'$ from $\Pi_2$;
    evaluate $\pi_1^a$ and $\pi_2'$ on $M$ and collect data;
    update $\theta^a$ via PG if enough data collected;
**until** *enough iterations*;
**return** $\pi_1^a(\theta^a)$;

---

remains the same for agent 2. Suppose a policy population $\mathcal{P} = \{\pi_2^{(1)}, \ldots, \pi_2^{(N)}\}$ is obtained during the RR phase, we first construct a diverse strategy set $\Pi_2 \subseteq \mathcal{P}$ that contains all the discovered behaviors from $\mathcal{P}$. Then we construct a mixed strategy by randomly sampling a policy $\pi_2'$ from $\Pi_2$ in every training episode and run PG to learn $\pi_1^a$ by competing against this constructed mixed strategy. The procedure is summarized in Algo. 2. Note that setting $\Pi_2 = \mathcal{P}$ appears to be a simple and natural choice. However, in practice, since $\mathcal{P}$ typically contains just a few strategic behaviors, it is unnecessary for $\Pi_2$ to include every individual policy from $\mathcal{P}$. Instead, it is sufficient to simply ensure $\Pi_2$ contains *at least* one policy from each equilibrium in $\mathcal{P}$ (more details in Sec. 5.3). Additionally, this method does not apply to the one-shot game setting (i.e., horizon is 1) because the adaptive agent does not have any prior knowledge about its opponent's identity before the game starts.

**Implementation:** We train an RNN policy for $\pi_1^a(\theta^a)$. It is critical that the policy input does not directly reveal the opponent's identity, so that it is forced to identify the opponent strategy through what it has observed. On the contrary, when adopting an actor-critic PG framework (Lowe et al., 2017), it is extremely beneficial to include the identity information in the critic input, which makes critic learning substantially easier and significantly stabilizes training. We also utilize a multi-head architecture adapted from the multi-task learning literature (Yu et al., 2019), i.e., use a separate value head for each training opponent, which empirically results in the best training performance.

## 4 Testbeds for RPG: Temporal Trust Dilemmas

We introduce three 2-player Markov games as testbeds for RPG. All these games have a diverse range of NE strategies including both "risky" cooperative NEs with high payoffs but hard to discover and "safe" non-cooperative NEs with lower payoffs. We call them *temporal trust dilemmas*. Game descriptions are in a high level to highlight the game dynamics. More details are in Sec. 5 and App. B.

**Gridworlds:** We consider two games adapted from Peysakhovich & Lerer (2018b), *Monster-Hunt* (Fig. 3) and *Escalation* (Fig. 4). Both games have a 5-by-5 grid and symmetric rewards.

*Monster-Hunt* contains a monster and two apples. Apples are static while the monster *keeps moving towards its closest agent*. If a single agent meets the monster, it *loses* a penalty of 2; if two agents catch the monster together, they both earn a bonus of 5. Eating an apple always raises a bonus of 2. Whenever an apple is eaten or the monster meets an agent, the entity will respawn randomly. The optimal payoff can only be achieved when both agents precisely catch the monster simultaneously.

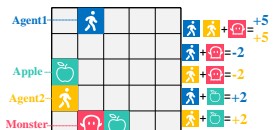

Figure 3: *Monster-Hunt*

*Escalation* contains a lit grid. When two agents both step on the lit grid, they both get a bonus of 1 and a neighboring grid will be lit up in the next timestep. If only one agent steps on the lit grid, it gets a penalty of $0.9L$, where $L$ denotes the consecutive cooperation steps until that timestep, and the lit grid will respawn randomly. Agents need to stay together on the lit grid to achieve the maximum payoff despite of the growing penalty. There are multiple NEs: for each $L$, that both agents cooperate for $L$ steps and then leave the lit grid jointly forms an NE.

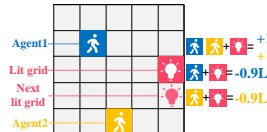

Figure 4: *Escalation*

*Agar.io* is a popular multiplayer online game. Players control cells in a Petri dish to gain as much mass as possible by eating smaller cells while avoiding being eaten by larger ones. Larger cells move slower. Each player starts with one cell but can split a sufficiently large cell into two, allowing them to control multiple cells (Wikipedia, 2020). We consider a simplified scenario (Fig. 5) with 2 players (agents) and tiny script cells, which automatically runs away when an agent comes by. There is a low-risk non-cooperative strategy, i.e., two agents stay away from each other and hunt script cells independently. Since the script cells move faster, it is challenging for a single agent to hunt them. By contrast, two agents can cooperate to encircle the script cells to accelerate hunting. However, cooperation is extremely risky for the agent with less mass: two agents need to stay close to cooperate but the larger agent may defect by eating the smaller one and gaining an immediate big bonus.

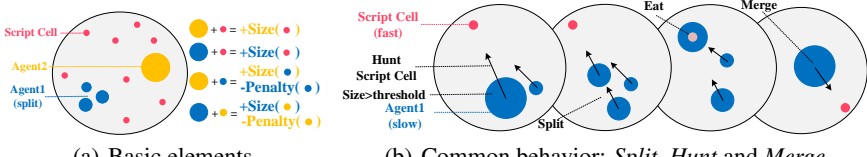

(a) Basic elements  (b) Common behavior: *Split*, *Hunt* and *Merge*

Figure 5: *Agar.io*: (a) a simplified 2-player setting; (b) basic motions: *split*, *hunt* script cells, *merge*.

## 5 EXPERIMENT RESULTS

In this section, we present empirical results showing that in all the introduced testbeds, including the real-world game *Agar.io*, RPG always discovers diverse strategic behaviors and achieves an equilibrium with substantially higher rewards than standard multi-agent PG methods. We use PPO (Schulman et al., 2017) for PG training. Training episodes for RPG are accumulated over all the perturbed games. Evaluation results are averaged over 100 episodes in gridworlds and 1000 episodes in *Agar.io*. We repeat all the experiments with 3 seeds and use $X_{(Y)}$ to denote mean $X$ with standard deviation $Y$ in all tables. Since all our discovered (approximate) NEs are symmetric for both players, we simply take $E(\pi_1, \pi_2) = U_1(\pi_1, \pi_2)$ as our evaluation function and only measure *the reward of agent 1* in all experiments for simplicity. More details can be found in appendix.

### 5.1 GRIDWORLD GAMES

**Monster-Hunt:** Each agent's reward is determined by three features per timestep: (1) whether two agents catch the monster together; (2) whether the agent steps on an apple; (3) whether the agent meets the monster alone. Hence, we write $\phi(s, a_1, a_2; i)$ as a 3-dimensional 0/1 vector with one dimension for one feature. The original game corresponds to $w = [5, 2, -2]$. We set $C_{\max} = 5$ for sampling $w$.

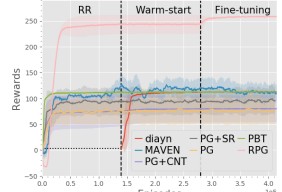

Figure 6: Full process of RPG in *Monster-Hunt*

We compare RPG with a collection of baselines, including standard PG (PG), PG with shared reward (PG+SR), population-based training (PBT), which trains the same amount of parallel PG policies as RPG, as

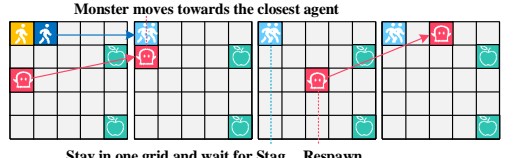 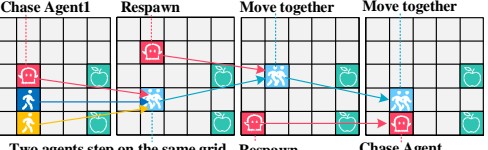

(a) Strategy w. $w=[5,0,0]$ and $w=[5,0,2]$ (by chance) | (b) The final strategy after fine-tuning

Figure 7: Emergent cooperative (approximate) NE strategies found by RPG in *Monster-Hunt*

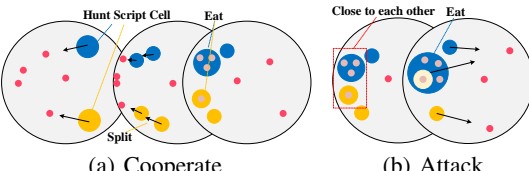

|        | PBT       | RR        | RPG       | RND       |
|--------|-----------|-----------|-----------|-----------|
| Rew.   | 3.8(0.3)  | 3.8(0.2)  | **4.3**(0.2) | 2.8(0.3)  |
| #Coop. | 1.9(0.2)  | **2.2**(0.1) | 2.0(0.3)  | 1.3(0.2)  |
| #Hunt  | 0.6(0.1)  | 0.4(0.0)  | **0.7**(0.0) | 0.6(0.1)  |

(a) Cooperate   (b) Attack

Figure 9: Emergent strategies in *standard Agar.io*: (a) agents *cooperate* to hunt efficiently; (b) a larger agent breaks the cooperation by *attacking* the other.

Table 2: Results in the *standard setting* of *Agar.io*. *PBT*: population training of parallel PG policies; *RR*: best policy in the RR phase ($w=[1,1]$); *RPG*: fine-tuned policy; *RND*: PG with RND bonus in the original game.

well as popular exploration methods, i.e., count-based exploration (PG+CNT) (Tang et al., 2017) and MAVEN (Mahajan et al., 2019). We also consider an additional baseline, DIAYN (Eysenbach et al., 2019), which discovers diverse skills using a *trajectory-based* diversity reward. For a fair comparison, we use DIAYN to first pretrain diverse policies (conceptually similar to the RR phase), then evaluate the rewards for every pair of obtained policies to select the best policy pair (i.e., evaluation phase, shown with the dashed line in Fig. 6), and finally fine-tune the selected policies until convergence (i.e., fine-tuning phase). The results of RPG and the 6 baselines are summarized in Fig. 6, where RPG consistently discovers a strategy with a significantly higher payoff. Note that the strategy with the optimal payoff may not always directly emerge in the RR phase, and there is neither a particular value of $w$ constantly being the best candidate: e.g., in the RR phase, $w = [5, 0, 2]$ frequently produces a sub-optimal cooperative strategy (Fig. 7(a)) with a reward lower than other $w$ values, but it can also occasionally lead to the optimal strategy (Fig. 7(b)). Whereas, with the fine-tuning phase, the overall procedure of RPG always produces the optimal solution. We visualize both two emergent cooperative strategies in Fig. 7: in the sub-optimal one (Fig. 7(a)), two agents simply move to grid (1,1) together, stay still and wait for the monster, while in the optimal one (Fig. 7(b)), two agents meet each other first and then actively move towards the monster jointly, which further improves hunting efficiency.

***Escalation***: We can represent $\phi(s, a_1, a_2; i)$ as 2-dimensional vector containing (1) whether two agents are both in the lit grid and (2) the total consecutive cooperation steps. The original game corresponds to $w = [1, -0.9]$. We set $C_{\max} = 5$ and show the total number of cooperation steps per episode for several selected $w$ values throughout training in Fig. 8, where RR is able to discover different NE strategies. Note that $w = [1, 0]$ has already produced the strategy with the optimal payoff in this game, so the fine-tuning phase is no longer needed.

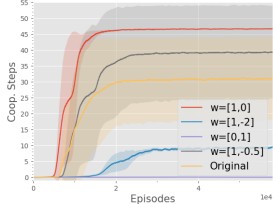

Figure 8: RR in *Escalation*

### 5.2 2-PLAYER GAMES IN *Agar.io*

There are two different settings of *Agar.io*: (1) the *standard setting*, i.e., an agent gets a penalty of $-x$ for losing a mass $x$, and (2) the more challenging *aggressive setting*, i.e., no penalty for mass loss. Note in both settings: (1) when an agent eats a mass $x$, it always gets a bonus of $x$; (2) if an agent loses all the mass, it immediately dies while the other agent can still play in the game. The aggressive setting promotes agent interactions and typically leads to more diverse strategies in practice. Since both settings strictly define the penalty function for mass loss, we do not randomize this reward term. Instead, we consider two other factors: (1) the bonus for eating the other *agent*; (2) the prosocial level of both agents. We use a 2-dimensional vector $w = [w_0, w_1]$, where $0 \leq w_0, w_1 \leq 1$, to denote a particular reward function such that (1) when eating a cell of mass $x$ from the other *agent*, the bonus is $w_0 \times x$, and (2) the final reward is a linear interpolation between $R(\cdot; i)$ and $0.5(R(\cdot; 0) + R(\cdot; 1))$ w.r.t. $w_1$, i.e., when $w_1 = 0$, each agent optimizes its individual reward while when $w_1 = 1$, two agents have a shared reward. The original game in both *Agar.io* settings corresponds to $w = [1, 0]$.

**Standard setting:** PG in the original game ($w = [1, 0]$) leads to a typical trust-dilemma dynamics: the two agents first learn to hunt and occasionally *Cooperate* (Fig. 9(a)), i.e., eat a script cell with the other agent close by; then accidentally one agent *Attacks* the other agent (Fig. 9(b)), which yields a big

|  | PBT | $w$=[0.5, 1] | $w$=[0, 1] | $w$=[0, 0] | RPG | RND |
|---|---|---|---|---|---|---|
| Rew. | 3.3(0.2) | 4.8(0.6) | 5.1(0.4) | 6.0(0.5) | **8.9**(0.3) | 3.2(0.2) |
| #Attack | 0.4(0.0) | 0.7(0.2) | 0.3(0.1) | 0.5(0.1) | **0.9**(0.1) | 0.4(0.0) |
| #Coop. | 0.0(0.0) | 0.6(0.6) | **2.3**(0.3) | 1.6(0.1) | 2.0(0.2) | 0.0(0.0) |
| #Hunt | 0.7(0.1) | 0.6(0.3) | 0.3(0.0) | 0.7(0.0) | **0.9**(0.1) | 0.7(0.0) |

Table 3: Results in the *aggressive setting* of *Agar.io*: *PBT*: population training of parallel PG policies; *RR*: $w$=[0, 0] is the best candidate via RR; *RPG*: fine-tuned policy; *RND*: PG with RND bonus.

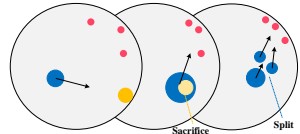

Figure 10: *Sacrifice* strategy, $w$=[1, 1], *aggressive setting*.

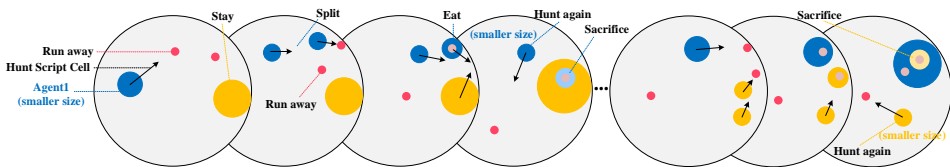

Figure 11: *Perpetual* strategy, $w$=[0.5, 1] (by chance), *aggressive setting*, i.e., two agents mutually sacrifice themselves. One agent first splits to sacrifice *a part* of its mass to the larger agent while the other agent also does the same thing later to repeat the sacrifice cycle.

immediate bonus and makes the policy aggressive; finally policies converge to the non-cooperative equilibrium where both agents keep apart and hunt alone. The quantitative results are shown in Tab. 2. Baselines include population-based training (PBT) and a state-the-art exploration method for high-dimensional state, *Random Network Distillation* (RND) (Burda et al., 2019). RND and PBT occasionally learns cooperative strategies while RR stably discovers a cooperative equilibrium with $w = [1, 1]$, and the full RPG further improves the rewards. Interestingly, the best strategy obtained in the RR phase even has a higher *Cooperate* frequency than the full RPG: fine-tuning transforms the strong cooperative strategy to a more efficient strategy, which has a better balance between *Cooperate* and selfish *Hunt* and produces a higher average reward.

**Aggressive setting:** Similarly, we apply RPG in the aggressive setting and show results in Tab. 3. Neither PBT nor RND was able to find any cooperative strategies in the aggressive game while RPG stably discovers a cooperative equilibrium with a significantly higher reward. We also observe a diverse set of complex strategies in addition to normal *Cooperate* and *Attack*. Fig. 10 visualizes the *Sacrifice* strategy derived with $w = [1, 1]$: the smaller agent rarely hunts script cells; instead, it waits in the corner for being eaten by the larger agent to contribute all its mass to its partner. Fig. 11 shows another surprisingly novel emergent strategy by $w = [0.5, 1]$: each agent first hunts individually to gain enough mass; then one agent splits into smaller cells while the other agent carefully eats ***a portion*** of the split agent; later on, when the agent who previously lost mass gains sufficient mass, the larger agent similarly splits itself to contribute to the other one, which completes the (ideally) never-ending loop of partial sacrifice. We name this strategy *Perpetual* for its conceptual similarity to the perpetual motion machine. Lastly, the best strategy is produced by $w = [0, 0]$ with a balance between *Cooperate* and *Perpetual*: they cooperate to hunt script cells to gain mass efficiently and quickly perform mutual sacrifice as long as their mass is sufficiently large for split-and-eat. Hence, although the RPG policy has relatively lower *Cooperate* frequency than the policy by $w = [0, 1]$, it yields a significantly higher reward thanks to a much higher *Attack* (i.e., *Sacrifice*) frequency.

## 5.3 LEARNING ADAPTIVE POLICIES

***Monster-Hunt***: We select policies trained by 8 different $w$ values in the RR phase and use half of them for training the adaptive policy and the remaining half as hidden opponents for evaluation. We also make sure that both training and evaluation policies cover the following 4 strategy modes: (1) *M(onster)*: the agent always moves towards the monster; (2) *M(onster)-Alone*: the agent moves towards the monster but also tries to keeps apart from the other agent;

| Oppo. | M. | M-Coop. | M-Alone. | Apple. |
|---|---|---|---|---|
| #C-H | 16.3(19.2) | 20.9(0.8) | 14.2(18.0) | 2.7(1.0) |
| #S-H | 1.2(0.4) | 0.4(0.1) | 2.2(1.2) | 2.2(1.4) |
| #Apple | 12.4(7.3) | 3.3(0.8) | 10.9(7.0) | 13.6(3.8) |

Table 4: Stats. of the adaptive agent in *Monster-Hunt* with *hold-out* test-time opponents. #C(oop.)-H(unt): both agents catch the monster; #S(ingle)-H(unt): the adaptive agent meets the monster alone; #Apple: apple eating. The adaptive policy successfully exploits different opponents and rarely meets the monster alone.

(3) *M(onster)-Coop.*: the agent seeks to hunt the monster together with the other agent; (4) *Apple*: the agent only eats apple. The evaluation results are shown in Tab. 4, where the adaptive policy successfully exploits all the test-time opponents, including *M(onster)-Alone*, which was trained to actively avoids the other agent.

***Agar.io***: We show the trained agent can choose to cooperate or compete adaptively in the standard setting. We pick 2 cooperative policies (i.e., *Cooperate* preferred, $w=[1,0]$) and 2 competitive policies (i.e., *Attack* preferred, $w=[1,1]$) and use half of them for training and the other half for testing. For a hard challenge at test time, we switch the opponent within an episode, i.e., we use a cooperative opponent in the first half and then immediately switch to a competitive one, and vice versa. So, a desired policy should adapt quickly at halftime. Tab. 5 compares the second-half behavior of the adaptive agent with the oracle pure-competitive/cooperative agents. The rewards of the adaptive agent is close to the oracle: even with half-way switches, the trained policy is able to exploit the cooperative opponent while avoid being exploited by the competitive one.

| Agent | Adapt. | Coop. | Comp. |
|---|---|---|---|
| Opponent: Cooperative $\rightarrow$ Competitive | | | |
| #Attack | 0.2(0.0) | 0.3(0.0) | 0.1(0.1) |
| Rew. | 0.7(0.7) | -0.2(0.6) | 0.8(0.5) |
| Opponent: Competitive $\rightarrow$ Cooperative | | | |
| #Coop. | 1.0(0.3) | 1.4(0.4) | 0.3(0.4) |
| Rew. | 2.5(0.7) | 3.6(1.2) | 1.1(0.7) |

Table 5: Adaptation test in *Agar.io*. Opponent type is switched half-way per episode. *#Attack*, *#Coop.*: episode statistics; *Rew.*: agent reward. Adaptive agents' rewards are close to oracles.

## 6 RELATED WORK AND DISCUSSIONS

Our core idea is reward perturbation. In game theory, this is aligned with the quantal response equilibrium (McKelvey & Palfrey, 1995), a smoothed version of NE obtained when payoffs are perturbed by a Gumbel noise. In RL, reward shaping is popular for learning desired behavior in various domains (Ng et al., 1999; Babes et al., 2008; Devlin & Kudenko, 2011), which inspires our idea for finding diverse *strategic* behavior. By contrast, state-space exploration methods (Pathak et al., 2017; Burda et al., 2019; Eysenbach et al., 2019; Sharma et al., 2020) only learn low-level primitives *without* strategy-level diversity (Baker et al., 2020).

RR trains a set of policies, which is aligned with the population-based training in MARL (Jaderberg et al., 2017; 2019; Vinyals et al., 2019; Long et al., 2020; Forestier et al., 2017). RR is conceptually related to domain randomization (Tobin et al., 2017) with the difference that we train separate policies instead of a single universal one, which suffers from mode collapse (see appendix D.2.3). RPG is also inspired by the map-elite algorithm (Cully et al., 2015) from evolutionary learning community, which optimizes multiple objectives simultaneously for sufficiently diverse polices. Our work is also related to Forestier et al. (2017), which learns a set of policies w.r.t. different fitness functions in the single-agent setting. However, they only consider a restricted fitness function class, i.e., the distance to each object in the environment, which can be viewed as a special case of our setting. Besides, RPG helps train adaptive policies against a set of opponents, which is related to Bayesian games (Dekel et al., 2004; Hartline et al., 2015). In RL, there are works on learning when to cooperate/compete (Littman, 2001; Peysakhovich & Lerer, 2018a; Kleiman-Weiner et al., 2016; Woodward et al., 2019; McKee et al., 2020), which is a special case of ours, or learning robust policies (Li et al., 2019; Shen & How, 2019; Hu et al., 2020), which complements our method.

Although we choose decentralized PG in this paper, RR can be combined with any other multi-agent learning algorithms for games, such as fictitious play (Robinson, 1951; Monderer & Shapley, 1996; Heinrich & Silver, 2016; Kamra et al., 2019; Han & Hu, 2019), double-oracle (McMahan et al., 2003; Lanctot et al., 2017; Wang et al., 2019; Balduzzi et al., 2019) and regularized self-play (Foerster et al., 2018; Perolat et al., 2020; Bai & Jin, 2020). Many of these works have theoretical guarantees to find an (approximate) NE but there is little work focusing on *which* NE strategy these algorithms can converge to when multiple NEs exist, e.g., the stag-hunt game and its variants, for which many learning dynamics fail to converge to a prevalence of the pure strategy Stag (Kandori et al., 1993; Ellison, 1993; Fang et al., 2002; Skyrms & Pemantle, 2009; Golman & Page, 2010)..

In this paper, we primarily focus on how reward randomization *empirically* helps MARL discover better strategies in practice and therefore only consider stag hunt as a particularly challenging example where an "optimal" NE with a high payoff for every agent exists. In general cases, we can select a desired strategy w.r.t. an evaluation function. This is related to the problem of equilibrium refinement (or equilibrium selection) (Selten, 1965; 1975; Myerson, 1978), which aims to find a subset of equilibria satisfying desirable properties, e.g., admissibility (Banks & Sobel, 1987), subgame perfection (Selten, 1965), Pareto efficiency (Bernheim et al., 1987) or robustness against opponent's deviation from best response in security-related applications (Fang et al., 2013; An et al., 2011).

ACKNOWLEDGMENTS

This work is supported by National Key R&D Program of China (2018YFB0105000). Co-author Fang is supported, in part, by a research grant from Lockheed Martin. Co-author Wang is supported, in part, by gifts from Qualcomm and TuSimple. The views and conclusions contained in this document are those of the authors and should not be interpreted as representing the official policies, either expressed or implied, of the funding agencies. The authors would like to thank Zhuo Jiang and Jiayu Chen for their support and input during this project. Finally, we particularly thank Bowen Baker for initial discussions and suggesting the Stag Hunt game as our research testbed, which eventually leads to this paper.

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

## A    PROOFS

*Proof of Theorem 1.*  We apply self-play policy gradient to optimize $\theta_1$ and $\theta_2$. Here we consider a projected version, i.e., if at some time $t$, $\theta_1$ or $\theta_2 \notin [0,1]$, we project it to $[0,1]$ to ensure it is a valid distribution.

We first compute the utility given a pair $(\theta_1, \theta_2)$

$$U_1(\theta_1, \theta_2) = a\theta_1\theta_2 + c\theta_1(1 - \theta_2) + b(1 - \theta_1)\theta_2 + d(1 - \theta_1)(1 - \theta_2)$$
$$U_2(\theta_1, \theta_2) = a\theta_1\theta_2 + b\theta_1(1 - \theta_2) + c(1 - \theta_1)\theta_2 + d(1 - \theta_1)(1 - \theta_2).$$

We can compute the policy gradient

$$\nabla U_1(\theta_1, \theta_2) = a\theta_2 + c(1 - \theta_2) - b\theta_2 - d(1 - \theta_2) = (a + d - b - c)\theta_2 + c - d$$
$$\nabla U_2(\theta_1, \theta_2) = a\theta_2 - b\theta_1 + c(1 - \theta_1) - d(1 - \theta_1) = (a + d - b - c)\theta_1 + c - d$$

Recall in order to find the optimal solution both $\theta_1$ and $\theta_2$ need to increase. Also note that the initial $\theta_1$ and $\theta_2$ determines the final solution. In particular, only if $\theta_1$ and $\theta_2$ are increasing at the beginning, they will converge to the desired solution.

To make *either* $\theta_1$ or $\theta_2$ increase, we need to have

$$(a + d - b - c)\theta_1 + c - d > 0 \text{ or } (a + d - b - c)\theta_2 + c - d > 0 \tag{1}$$

Consider the scenario $a - b = \epsilon(d - c)$. In order to make Inequality equation 1 to hold, we need at least either $\theta_1, \theta_2 \geq \frac{1}{1+\epsilon}$.

If we initialize $\theta_1 \sim [0,1]$ and $\theta_2 \sim [0,1]$, the probability of either $\theta_1, \theta_2 \geq \frac{1}{1+\epsilon}$ is $1 - \left(\frac{1}{1+\epsilon}\right)^2 = \frac{2\epsilon+\epsilon^2}{1+2\epsilon+\epsilon^2} = O(\epsilon)$. ∎

*Proof of Theorem 2.*  Using a similar observation as in Theorem 1, we know a *necessary* condition to make PG converge to a sub-optimal NE is

$$(a + d - b - c)\theta_1 + c - d < 0 \text{ or } (a + d - b - c)\theta_2 + c - d < 0.$$

Based on our generating scheme on $a, b, c, d$ and the initialization scheme on $\theta_1, \theta_2$, we can verify that Therefore, via a union bound, we know

$$\mathbb{P}\left((a + d - b - c)\theta_1 + c - d < 0 \text{ or } (a + d - b - c)\theta_2 + c - d < 0\right) \leq 0.6. \tag{2}$$

Since each round is independent, the probability that PG fails for all $N$ times is upper bounded by $0.6^N$. Therefore, the success probability is lower bounded by $1 - 0.6^N = 1 - \exp\left(-\Omega(N)\right)$.

∎

## B    ENVIRONMENT DETAILS

### B.1    *Iterative Stag-Hunt*

In *Iterative Stag-Hunt*, two agents play 10 rounds, that is, both PPO's trajectory length and episode length are 10. Action of each agent is a 1-dimensional vector, $a_i = \{t_i, i \in \{0, 1\}\}$, where $t_i = 0$ denotes taking *Stag* action and $t_i = 1$ denotes taking *Hare* action. Observation of each agent is actions taking by itself and its opponent in the last round, i.e., $o_i^r = \{a_i^{r-1}, a_{1-i}^{r-1}; i \in \{0, 1\}\}$, where $r$ denotes the playing round. Note that neither agent has taken action at the first round, so the observation $o_i = \{-1, -1\}$.

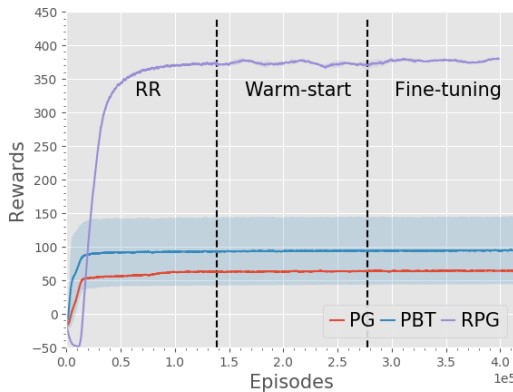

Figure 12: Results on *Monster-Hunt* with 3 agents (3 seeds).

### B.2   *Monster-Hunt*

In *Monster-Hunt*, two agents can move one step in any of the four cardinal directions (*Up*, *Down*, *Left*, *Right*) at each timestep. Let $a_i = \{t_i, i \in \{0, 1\}\}$ denote action of agent $i$, where $t_i$ is a discrete 4-dimensional one-hot vector. The position of each agent can not exceed the border of 5-by-5 grid, where action execution is invalid. One Monster and two apples respawn in the different grids at the initialization. If an agent eats (move over in the grid world) an apple, it can gain 2 points. Sometimes, two agents may try to eat the same apple, the points will be randomly assigned to only one agent. Catching the monster alone causes an agent **lose** 2 points, but if two agents catch the stag simultaneously, each agent can gain 5 points. At each time step, the monster and apples will respawn randomly elsewhere in the grid world if they are wiped. In addition, the monster chases the agent closest to it at each timestep. The monster may move over the apple during the chase, in this case, the agent will gain the sum of points if it catches the monster and the apple exactly. Each agent's observation $o_i$ is a 10-dimensional vector and formed by concatenating its own position $p_i$, the other agent's position $p_{1-i}$, monster's position $p_{monster}$ and sorted apples' position $p_{apple0}, p_{apple1}$, i.e., $o_i = \{p_i, p_{1-i}, p_{monster}, p_{apple0}, p_{apple1}; i \in \{0, 1\}\}$, where $p = (u, v)$ denotes the 2-dimensional coordinates in the gridworld.

### B.3   *Monster-Hunt* WITH MORE THAN 2 AGENTS

Here we consider extending RPG to the general setting of $N$ agents. In most of the multi-agent games, the reward function are fully symmetric for the same type of agents. Hence, as long as we can formulate the reward function in a linear form over a feature vector and a shared weight, i.e., $R(s, a_1, \ldots, a_N; i) = \phi(s, a_1, \ldots, a_N; i)^T w$, we can directly apply RPG without any modification by setting $\mathcal{R} = \{R_w : R_w(s, a_1, \ldots, a_N; i) = \phi(s, a_1, \ldots, a_N; i)^T w\}$. Note that typically the dimension of the feature vector $\phi(\cdot)$ remains fixed w.r.t. different number of agents ($N$). For example, in the *Agar.io* game, no matter how many players are there in the game, the rule of how to get reward bonus and penalties remains the same.

Here, we experiment RPG in *Monster-Hunt* with 3 agents. The results are shown in Fig. 12. We consider baselines including the standard PG (PG) and population-based training (PBT). RPG reliably discovers a strong cooperation strategy with a substantially higher reward than the baselines.

### B.4   *Escalation*

In *Escalation*, two agents appear randomly and one grid lights up at the initialization. If two agents step on the lit grid simultaneously, each agent can gain 1 point, and the lit grid will go out with an adjacent grid lighting up. Both agents can gain 1 point again if they step on the next lit grid together. But if one agent steps off the path, the other agent will **lose** $0.9L$ points, where $L$ is the current length of stepping together, and the game is over. Another option is that two agents choose to step off the path simultaneously, neither agent will be punished, and the game continues. As the length $L$ of stepping together increases, the cost of betrayal increases linearly. $a_i = \{t_i, i \in \{0, 1\}\}$ denotes

action of agent $i$, where $t_i$ is a discrete 4-dimensional one-hot vector. The observation $a_i$ of agent $i$ is composed of its own position $p_i$, the other agent's position $p_{1-i}$ and the lit grid's position $p_{lit}$, i.e., $o_i = \{p_i, p_{1-i}, p_{lit}; i \in \{0, 1\}\}$, where $p = (u, v)$ denotes the 2-dimensional coordinates in the gridworld. Moreover, we utilize GRU to encode the length $L$ implicitly, instead of observing that explicitly.

### B.5 *Agar.io*

In the original online game *Agar.io*, multiple players are limited in a circle petri dish. Each player controls one or more balls using only a cursor and 2 keyboard keys "space" and "w". all balls belonging to the player will move forward to where the cursor pointing at. Balls larger than a threshold will split to 2 smaller balls and rush ahead when the player pressing the key "space". Balls larger than another threshold will emit tiny motionless food-like balls when the player pressing "w". *Agar.io* has many play modes like "Free-For-All" mode (All players fight for their own and can eat each other) and "Team" mode (Players are separated to two groups. They should cooperate with other players in the same group and eat other players belonging to another group).

We simplified settings of the original game *Agar.io*: Now agents don't need to emit tiny motionless balls and all fight with each other (FFA mode). The **action space** of the game is $target \times \{split, no\_split\}$. $target \in [0, 1]^2$ means the target position that all balls belonging to the agent move to. binary action $split$ or $no\_split$ means whether the player chooses to split, which will cause all balls larger than a threshold split to 2 smaller ones and rush ahead for a short while. These split balls will re-merge after some time, then the agent can split again. When one agent's ball meets another agent's ball and the former one is at least 1.2 times larger than the later, the later will be eaten and the former will get all its mass. The reward is defined as the increment of balls' mass. So every agent's goal is getting larger by eating others while avoid being eaten. But larger ball moves slower. So it's really hard to catch smaller balls only by chasing after it. Split will help, but it needs high accuracy to rush to the proper direction. In our experiments, there were 7 agents interacting with each other. 2 agents were learned by our algorithm and would quit the game if all balls were eaten. 5 agents were controlled by a script and would reborn at a random place if all balls were eaten. Learn-based agents were initialized larger than script-based agents so it was basically one-way catching. In this setting, cooperation was the most efficient behavior for learn-based agents to gain positive reward, where they coordinated to surround script-based agents and caught them.

**Observation space**: We denote partial observation of agent $i$ as $o_i$, which includes global information of the agent (denoted as $o_{i,global}$) and descriptions of all balls around the agent (including balls owned by the agent, denoted as $o_{i,balls}$. and $o_{i,balls} = \{o_{i,ball,1}, o_{i,ball,2}, ..., o_{i,ball,m}\}$, where $o_{i,ball,j}$ denotes the j-th ball around the agent and there are $m$ observed balls in all). $o_{i,global} = \{l_{i,obs}, w_{i,obs}, p_{i,center}, v_i, s_{i,alive}, n_{i,own}, n_{i,script}, n_{i,other}, a_{i,last}, r_{i,max}, r_{i,min}, m_i\}$ where $l_{i,obs}, w_{i,obs}$ (they are both 1D filled with a real number, from here the form like (1D, real) will be used as the abbreviation) are the length and width of the agent's observation scope, $p_{i,center}$ (2D, real) is its center position, $v_i$ (2D, real) is the speed of its center, $s_{i,alive}$(1D, binary) is whether the other learn-based agent is killed, $n_{i,own}, n_{i,script}, n_{i,other}$(1D, real) are numbers of each type of balls nearby (3 types: belonging to me, or belonging to a script agent, or belonging to another learn-based agent), $a_{i,last}$(3D, real) is the agent's last action, $r_{i,max}, r_{i,min}$(1D, real) are maximal and minimal radius of all balls belonging to the agent. for any $j = 1, 2, ..., m$, $o_{i,ball,j} = \{p_{i,j,relative}, p_{i,j,absolute}, v_{i,j}, v_{i,j,rush}, r_{i,j}, log(r_{i,j}), d_{i,j}, e_{i,j,max}, e_{i,j,min}, s_{i,j,rem}, t_{i,j}\}$, where $p_{i,j,relative}, p_{i,j,absolute}$(2D, real) are the ball's relative and absolute position, $v_{i,j}$ is its speed, $v_{i,j,rush}$ is the ball's additional rushing speed(when a ball splits to 2 smaller balls, these 2 balls will get additional speed and it's called $v_{i,j,rush}$, otherwise $v_{i,j,rush} = \mathbf{0}$), $r_{i,j}$(1D, real) is its radius, $d_{i,j}$ is the distance between the ball and the center of the agent, $e_{i,j,max}, e_{i,j,min}$(1D, binary) are whether the ball can be eaten by the maximal or minimal balls of the observing agent, $s_{i,j,rem}$(1D, binary) is whether the ball is able to remerge at present. $t_{i,j}$(3D, one hot) is the type of the ball.

The script-base agent can automatically chase after and split towards other smaller agents. When facing *extreme danger* (we define "*extreme danger*" as larger learn-based agents being very close to it), it will use a 3-step deep-first-search to plan a best way for escape. More details of the script can be seen in our code. We played against the script-base agent using human intelligence for many times and we could never hunt it when having only one ball and rarely catch it by split.

## C TRAINING DETAILS

### C.1 GRIDWORLD GAMES

In *Monster-Hunt* and *Escalation*, agents' networks are organized by actor-critic (policy-value) architecture. We consider $N = 2$ agents with a policy profile $\pi = \{\pi_0, \pi_1\}$ parameterized by $\theta = \{\theta_0, \theta_1\}$. The policy network $\pi_i$ takes observation $o_i$ as input, two hidden layers with 64 units are followed after that, and then outputs action $a_i$. While the value network takes as input observations of two agents, $o = \{o_0, o_1\}$ and outputs the V-value of agent $i$, similarly two hidden layers with 64 units are added before the output.

In *Escalation*, we also place an additional GRU module before the output in policy network and value network respectively, to infer opponent's intentions from historical information. Note that 64-dimensional hidden state of GRU $h$ will change if the policy network is updated. In order to both keep forward information and use backward information to compute generalized advantage estimate (GAE) with enough trajectories, we split buffer data into small chunks, e.g., 10 consecutive timesteps as a small data chunk. The initial hidden state $h_{init}$, which is the first hidden state $h_0$, is kept for each data chunk, but do another forward pass to re-compute $\{h_1, ..., h_{M-1}\}$, where $M$ represents the length of one data chunk, and keep buffer-reuse low, e.g., 4 in practice.

Agents in *Monster-Hunt* and *Escalation* are trained by PPO with independent parameters. Adam optimizer is used to update network parameters and each experiment is executed for 3 times with random seeds. More optimization hyper-parameter settings are in Tab.6. In addition, *Monster-Hunt* also utilizes GRU modules to infer opponent's identity during adaption training and the parallel threads are set to 64.

**Count-based exploration:** We just add the count-based exploration intrinsic reward $r_{int}$ to the environment reward during training. when the agent's observation is $o$, $r_{int} = \alpha/n_o$ where $\alpha$ is a hyperparameter adjusted properly (0.3 in *Monster-Hunt* and 1 in *Escalation*) and $n_o$ is the number of times the agent have the observation $o$.

**DIAYN:** In *Monster-Hunt*, we use DIAYN to train 10 diverse policy in the first 140k episodes (DIAYN's discriminator has 3 FC layers with 256, 128, 10 units respectively) and choose the policy which has the best performance in *Monster-Hunt*'s reward settings to fine-tune in the next 280k episodes. Note that DIAYN doesn't have a warm-start phase before fine-tuning in its original paper so we didn't do so as well. Note that in the first unsupervised learning phase, DIAYN does not optimize for any specific reward function. Hence, we did not plot the reward curve for DIAYN in Fig.7 for this phase. Instead, we simply put a dashed line showing the reward of the best selected pair of policies from DIAYN pretraining.

**MAVEN:** We use the open-sourced implementation of MAVEN from `https://github.com/AnujMahajanOxf/MAVEN`.

**Population-based training:** In each PBT trial, we straightforward train the same amount of parallel PG policies as RPG with different random seeds in each problem respectively and choose the one with best performance as the final policy. Note that the final training curve is averaged over 3 PBT trials.

### C.2 *Agar.io*

In *Agar.io*, we used PPO as our algorithm and agents' networks were also organized by actor-critic (policy-value) architecture with a GRU unit (i.e., PPO-GRU). We consider $N = 2$ agents with a policy profile $\pi = \{\pi_0, \pi_1\}$ sharing parameter $\theta$. The policy network $\pi_i$ takes observation $o_i$ as input. At the beginning, like (Baker et al., 2019), $o_{i,balls}$ is separated to 3 groups according to balls' types: $o_{i,ownballs}$, $o_{i,scriptballs}$ and $o_{i,otherballs}$. 3 different multi-head attention models with 4 heads and 64 units for transformation of keys, inquiries and values are used to embed information of 3 types of balls respectively, taking corresponding part of $o_{i,balls}$ as values and inquiries and $o_{i,global}$ as keys. Then their outputs are concatenated and transformed by an FC layer with 128 units before being sent to a GRU block with 128 units. After that, the hidden state is copied to 2 heads for policy's and value's output. The policy head starts with 2 FC layers both with 128 units and ends with 2 heads to generate discrete(*split* or *no_split*) and continuous(*target*) actions. The value head has 3 FC layers with 128, 128, 1 unit respectively and outputs a real number.

| Hyper-parameters | Value |
|---|---|
| Initial learning rate | 1e-3 |
| Minibatch size | 320 chunks of 10 timesteps |
| Adam stepsize ($\epsilon$) | 1e-5 |
| Discount rate ($\gamma$) | 0.99 |
| GAE parameter ($\lambda$) | 0.95 |
| Value loss coefficient | 1 |
| Entropy coefficient | 0.01 |
| Gradient clipping | 0.5 |
| PPO clipping parameter | 0.2 |
| Parallel threads | 64(*Escalation*),256(*Monster-Hunt*) |
| PPO epochs | 4 |
| reward scale parameter | 0.1 |
| episode length | 50 |

Table 6: PPO hyper-parameters used in Gridworld games, learning rate is linearly annealed during training.

| Hyper-parameters | Value |
|---|---|
| Learning rate | 2.5e-4 |
| Minibatch size | 2 * 512 chunks of 32 timesteps |
| Adam stepsize ($\epsilon$) | 1e-5 |
| Discount rate ($\gamma$) | 0.995 |
| GAE parameter ($\lambda$) | 0.95 |
| Value loss coefficient | 0.5 |
| action loss coefficient | 1 |
| Entropy coefficient | 0.01(discrete), 0.0025(continuous) |
| Gradient clipping | 20 |
| PPO clipping parameter | 0.1 |
| Parallel threads | 128 |
| PPO epochs | 4 |
| episode length | 128 |

Table 7: PPO hyper-parameters used in *Agar.io*

PPO-GRU was trained with 128 parallel environment threads. *Agar.io*'s episode length was uniform-randomly sampled between 300 and 400 both when training and evaluating. Buffer data were split to small chunks with $length = 32$ in order to diversify training data and stabilize training process. and the buffer was reused for 4 times to increase data efficiency. Hidden states of each chunk except at the beginning were re-computed after each reuse to sustain PPO's "on-policy" property as much as possible. Action was repeated for 5 times in the environment whenever the policy was executed and only the observation after the last action repeat was sent to the policy. Each training process started with a curriculum-learning in the first $1.5e7$ steps: Speed of script agents was multiplied with $x$, where $x$ is uniformly random-sampled between $max\{0, (n - 1e7)/5e6\}$ and $min\{1, max\{0, (n - 5e6)/5e6\}\}$ at the beginning of each episode, where $n$ was the steps of training. After the curriculum learning, Speed was fixed to the standard. Each experiment was executed for 3 times with different random seeds. Adam optimizer was used to update network parameters. More optimization hyper-parameter settings are in Tab.7.

# D  ADDITIONAL EXPERIMENT RESULTS

## D.1  *Monster-Hunt*

In *Monster-Hunt*, we set $C_{\max} = 5$ for sampling $w$. Fig. 13 illustrates the policies discovered by several selected $w$ values, where different strategic modalities can be clearly observed: e.g., with $w = [0, 5, 0]$, agents always avoid monsters and only eat apples. In Fig. 14, it's worth noting that $w = [5, 0, 2]$ could yield the best policy profile (i.e., two agents move together to hunt the monster.)

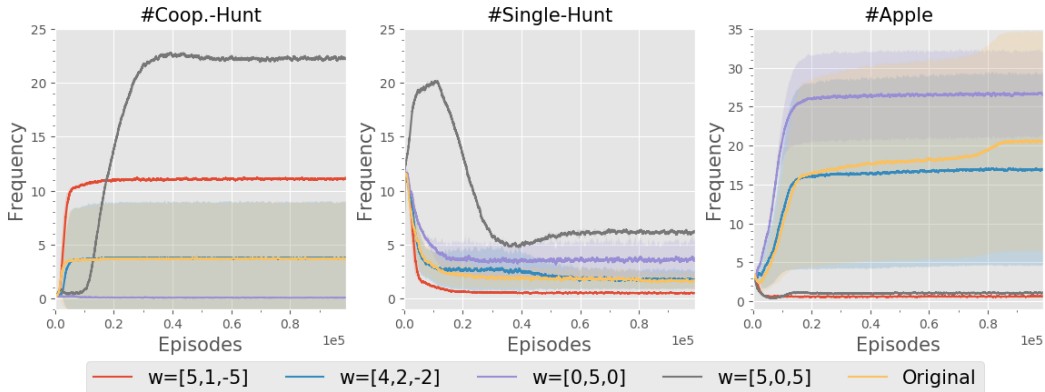

Figure 13: Statistics of different policy profiles in *Monster-Hunt*.#Coop.-Hunt: frequency of both agents catching the monster; #Single-Hunt: frequency of agents meeting the monster alone; #Apple: apple frequency.

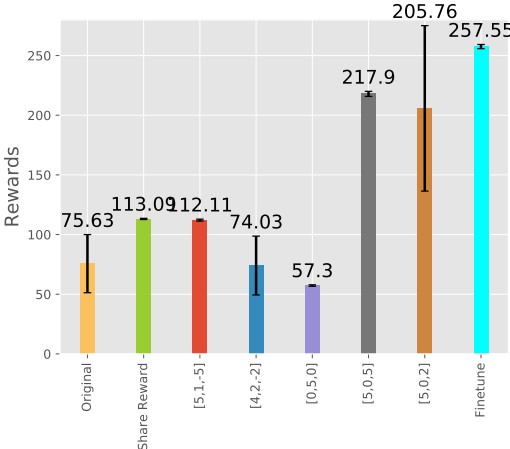

Figure 14: Results in original *Monster-Hunt*. Original: PG in the original game; Share reward: PG with shared reward in the original game; Finetune: fine-tuning the best policy obtained in the RR phase and yielding the highest reward in the original game.

and doesn't even require further fine-tuning with some seeds. But the performance of $w = [5, 0, 2]$ is significantly unstable and it may converge to another NE (i.e., two agents move to a corner and wait for the monster.) with other seeds. So $w = [5, 0, 5]$, which yields stable strong cooperation strategies with different seeds, will be chosen in RR phase when $w = [5, 0, 2]$ performs poorly. We demonstrate the obtained rewards from different policies in Fig. 14, where the policies learned by RPG produces the highest rewards.

## D.2 *Agar.io*

### D.2.1 STANDARD SETTING

We sampled 4 different $w$ and they varied in different degrees of cooperation. We also did experiments using only baseline PG or PG with intrinsic reward generated by *Random Network distillation* (RND) to compare with RPG. RR lasted for 40M steps, but only the best reward parameter in RR ($w = [1, 1]$) was warmed up for 3M steps and fine-tuned for 17M steps later. PG and RND were also trained for 60M steps in order to compare with RPGfairly. In Fig. 15, we can see that PG and RND produced very low rewards because they all converged to non-cooperative policies. $w = [1, 1]$ produced highest rewards after **RR**, and rewards boosted higher after fine-tuning.

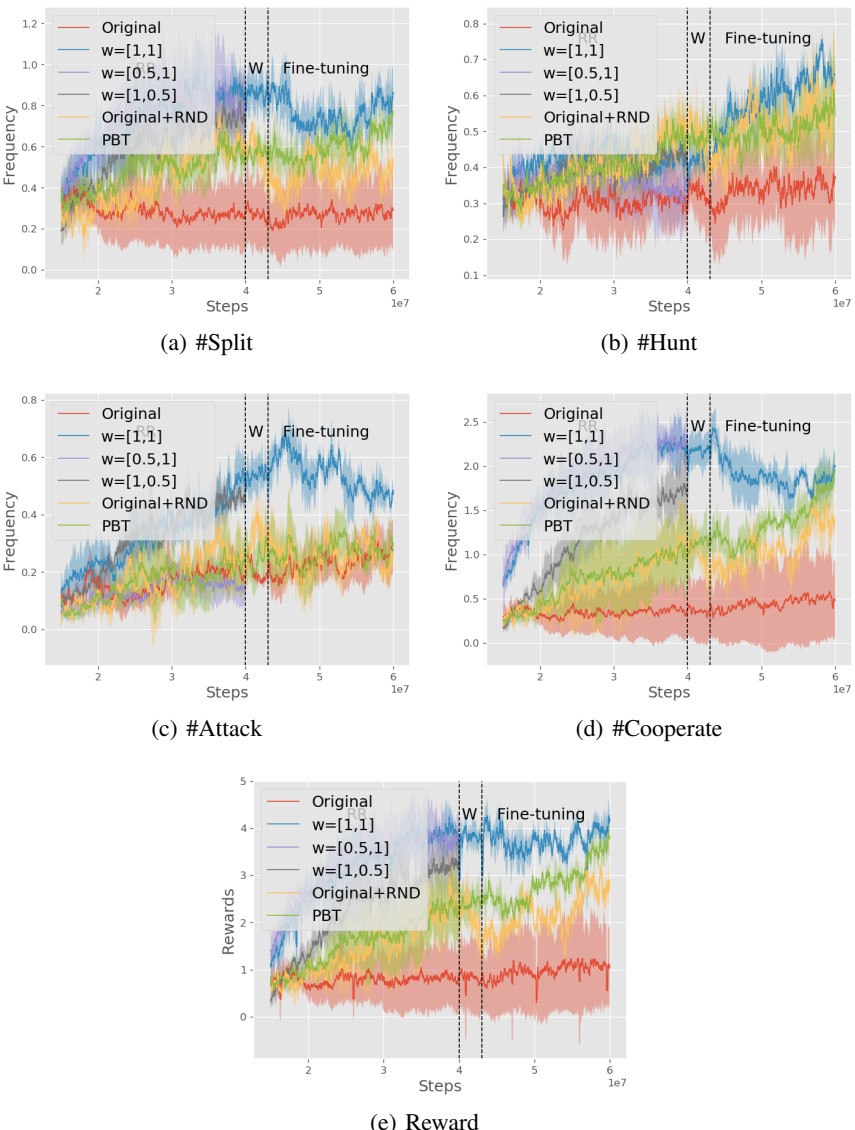

Figure 15: statistics of *standard setting* of *Agar.io*. (a) to (d) illustrate frequencies of *Split*, *Hunt*, *Attack* and *Cooperate* during training under different reward parameters and algorithms. *Split* means catching a script agent ball by splitting, *Hunt* means catching a script agent ball without splitting, *Attack* means catching a learn-based agent ball, *Cooperate* means catching a script agent ball while the other learn-based agent is close by.(the same below) (e) illustrates rewards of different policies.

### D.2.2  AGGRESSIVE SETTING

We sampled 5 different $w$ and their behavior were much more various. the other training settings were the same as *standard setting*. in Fig. 16, we should notice that simply sharing reward ($w = [1, 1]$) didn't get very high reward because attacking each other also benefits each other, so 2 agents just learned to sacrifice, Again, Fig. 16 illustrates that rewards of RPG was far ahead the other policies while both PG and PG+RND failed to learn cooperative strategies.

We also listed all results of *Standard* and *Aggressive setting* in Tab. 8 for clearer comparison.

### D.2.3  UNIVERSAL REWARD-CONDITIONED POLICY

We also tried to train a universal policy conditioned on $w$ by randomly sampling different $w$ at the beginning of each episode during training rather than fixing different $w$ and training the policy later

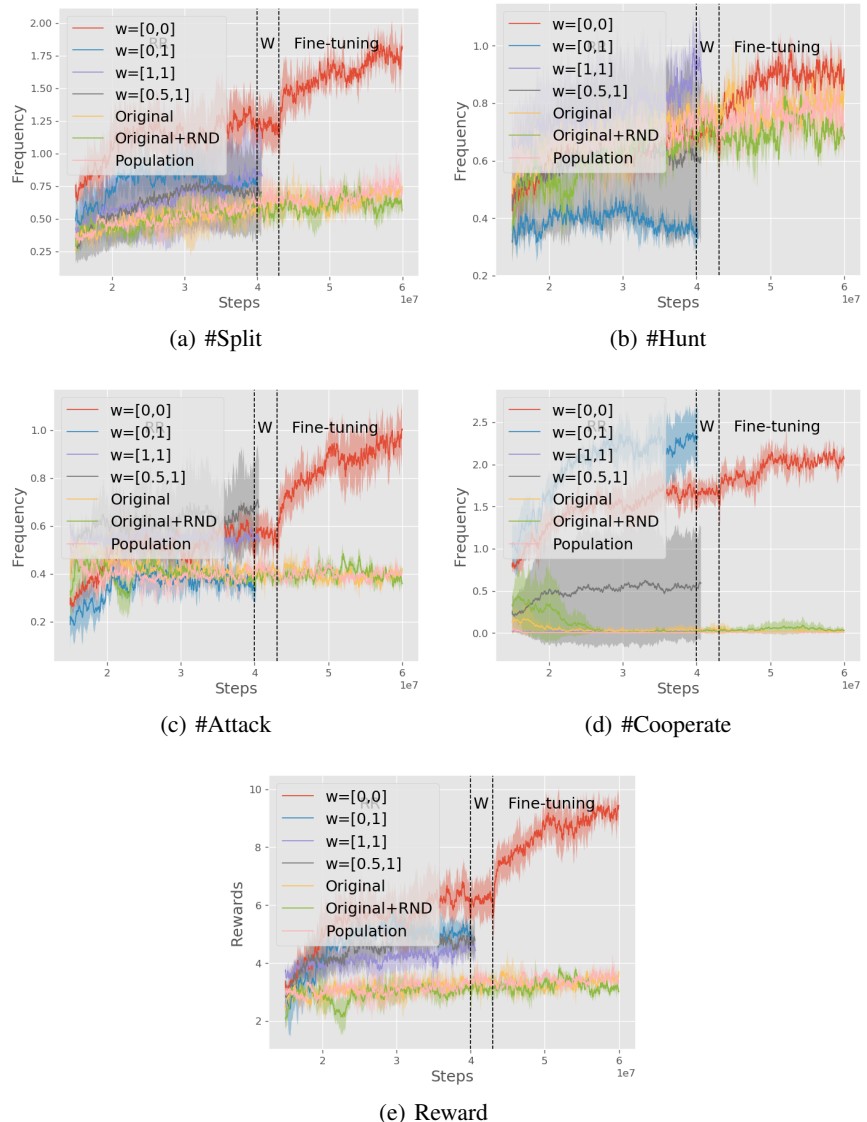

Figure 16: Statistics of *aggressive setting* of *Agar.io*. (a) to (d) illustrate frequencies of *Split*, *Hunt*, *Attack* and *Cooperate* during training under different reward parameters and algorithms.(e) illustrates rewards of different policies.

on. But as Fig. 17 illustrates, the learning process was very unstable and model performed almost the same under different $w$ due to the intrinsic disadvantage of an on-policy algorithm dealing with multi-tasks: the learning algorithm may pay more effort on $w$ where higher rewards are easier to get but ignore the performance on other $w$, which made it very hard to get diverse behaviors.

### D.3 LEARN ADAPTIVE POLICY

In this section, we add the opponents' identity $\psi$ in the input of the value network to stable the training process and boost the performance of the adaptive agent. $\psi$ is a $C$-dimensional one-hot vector, where $C$ denotes the number of opponents.

#### D.3.1 *Iterative Stag-Hunt*

In *Iterative Stag-Hunt*, we randomize the payoff matrix, which is a 4-dimensional vector, and set $C_{max} = 4$ for sampling $w$. The parallel threads are 512 and the episode length is 10. Other training hyper-parameter settings are the same as Tab.6. Fig 18 describes different $w = [a, b, c, d]$ (i.e.,

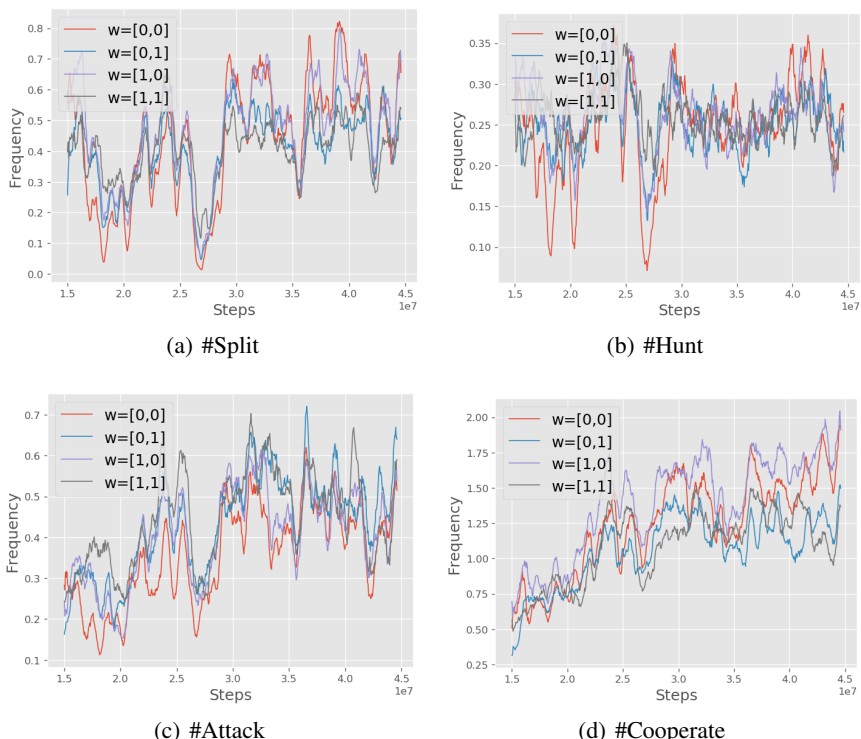

(a) #Split

(b) #Hunt

(c) #Attack

(d) #Cooperate

Figure 17: Statistics of Universal policy of *Agar.io*. (a) to (d) illustrate the frequency of *Split*, *Hunt*, *Attack* and *Cooperate* when fixing different $w$ while evaluating.

| Settings | Policy | Rewards | #Split | #Hunt | #Attack | #Cooperate |
|---|---|---|---|---|---|---|
| Standard | $w$=[1,1] | $3.843_{(0.23)}$ | $0.859_{(0.083)}$ | $0.411_{(0.034)}$ | $0.526_{(0.064)}$ | $2.203_{(0.136)}$ |
| | RPG | $4.34_{(0.171)}$ | $0.971_{(0.13)}$ | $0.659_{(0.048)}$ | $0.548_{(0.038)}$ | $2.028_{(0.297)}$ |
| | $w$=[0.5,1] | $3.827_{(0.489)}$ | $0.807_{(0.192)}$ | $0.365_{(0.106)}$ | $0.15_{(0.064)}$ | $2.342_{(0.286)}$ |
| | $w$=[1,0.5] | $3.174_{(0.653)}$ | $0.718_{(0.148)}$ | $0.432_{(0.026)}$ | $0.458_{(0.031)}$ | $1.716_{(0.418)}$ |
| | Original | $1.08_{(0.836)}$ | $0.3_{(0.19)}$ | $0.361_{(0.134)}$ | $0.291_{(0.098)}$ | $0.483_{(0.442)}$ |
| | RND | $2.789_{(0.346)}$ | $0.499_{(0.061)}$ | $0.623_{(0.128)}$ | $0.242_{(0.037)}$ | $1.349_{(0.164)}$ |
| | PBT | $3.822_{(0.347)}$ | $0.744_{(0.129)}$ | $0.585_{(0.146)}$ | $0.297_{(0.055)}$ | $1.935_{(0.167)}$ |
| Aggressive | $w$=[0,0] | $5.966_{(0.539)}$ | $1.195_{(0.155)}$ | $0.699_{(0.008)}$ | $0.517_{(0.066)}$ | $1.603_{(0.127)}$ |
| | RPG | $8.907_{(0.292)}$ | $1.655_{(0.138)}$ | $0.862_{(0.053)}$ | $0.903_{(0.081)}$ | $2.039_{(0.209)}$ |
| | $w$=[0,1] | $5.066_{(0.375)}$ | $0.785_{(0.041)}$ | $0.344_{(0.049)}$ | $0.346_{(0.058)}$ | $2.327_{(0.311)}$ |
| | $w$=[1,1] | $4.622_{(0.277)}$ | $0.836_{(0.304)}$ | $0.934_{(0.108)}$ | $0.552_{(0.019)}$ | $0.028_{(0.023)}$ |
| | $w$=[0.5,1] | $4.79_{(0.588)}$ | $0.678_{(0.31)}$ | $0.617_{(0.28)}$ | $0.67_{(0.194)}$ | $0.55_{(0.643)}$ |
| | Original | $3.551_{(0.121)}$ | $0.717_{(0.032)}$ | $0.812_{(0.078)}$ | $0.412_{(0.018)}$ | $0.027_{(0.026)}$ |
| | RND | $3.189_{(0.154)}$ | $0.626_{(0.065)}$ | $0.705_{(0.008)}$ | $0.382_{(0.029)}$ | $0.035_{(0.027)}$ |
| | PBT | $3.348_{(0.222)}$ | $0.697_{(0.133)}$ | $0.732_{(0.096)}$ | $0.396_{(0.014)}$ | $0.007_{(0.005)}$ |

Table 8: Frequencies of 4 types of events and rewards of different policies of *Agar.io* after completely training.

$[4, 0, 0, 0], [0, 0, 0, 4], [0, 4, 4, 0], [4, 1, 4, 0]$) yields different policy profiles. e.g., with $w = [0, 0, 0, 4]$, both agents tend to eat the hare. The original game corresponds to $w = [4, 3, -50, 1]$. Tab. 9 reveals $w = [4, 0, 0, 0]$ yields the highest reward and reaches the optimal NE without further fine-tuning.

| | Original | $w = [4, 0, 0, 0]$ | $w = [0, 0, 0, 4]$ | $w = [0, 4, 4, 0]$ | $w = [4, 1, 4, 0]$ |
|---|---|---|---|---|---|
| #Rewards | $20.00_{(0.00)}$ | $74.76_{(2.88)}$ | $20.00_{(0.00)}$ | $-470.0_{(0.00)}$ | $-453.45_{(0.25)}$ |

Table 9: Evaluation of different policy profiles obtained via RR in original *Iterative Stag-Hunt*. Note that $w = [4, 0, 0, 0]$ has the best performance among the policy profiles, and is the optimal NE with no further fine-tuning.

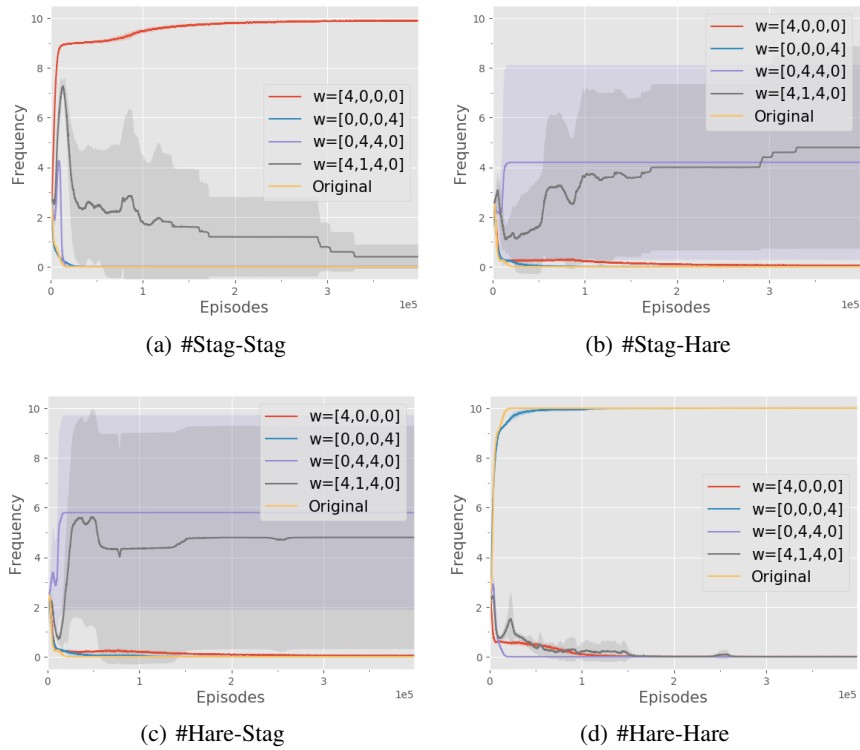

(a) #Stag-Stag

(b) #Stag-Hare

(c) #Hare-Stag

(d) #Hare-Hare

Figure 18: Find different policy profiles via Reward Randomization in *Iterative Stag-Hunt*. #Stag-Stag: the frequency of two agents both hunt the stag. #Stag-Hare: the frequency of agent1 hunts the stag while agent2 eats the hare. #Hare-Stag: the frequency of agent1 eat the hare while agent2 hunts the stag. #Hare-Hare: the frequency of two agents both eat the hare. Frequency: times of certain behavior performed in one episode.

| Oppo. Type | Stag | Hare | TFT | Random |
|---|---|---|---|---|
| #Stag | 9.31(0.77) | 3.6(4.33) | 7.31(3.82) | 5.35(3.48) |
| #Hare | 0.69(0.77) | 6.4(4.33) | 2.69(3.81) | 4.65(3.48) |

Table 10: Statistics of the adaptive policy in *Iterative Stag-Hunt* with 4 hand-designed opponents with different behavior preferences. #Stag: the adaptive agent hunts the stag; #Hare: the adaptive agent eats the hare; The adaptive policy successfully exploits different opponents, including cooperating with TFT opponent, which is totally different from trained opponents.

Utilizing 4 different strategies obtained in the RR phase as opponents, we could train an adaptive policy which can make proper decisions according to opponent's identity. Fig. 19 shows the adaption training curve, we can see that the policy yields adaptive actions stably after $5e4$ episodes. At the evaluation stage, we introduce 4 hand-designed opponents to test the performance of the adaptive policy, including *Stag* opponent (i.e., always hunt the stag), *Hare* opponent (i.e., always eat the hare), *Tit-for-Tat (TFT)* opponent (i.e., always hunt the stag at the first step, and then take the action executed by the other agent in the last step), and *Random* opponent (i.e., randomly choose to hunt the stag or eat the hare at each step). Tab. 10 illustrates that the adaptive policy exploits all hand-designed strategies, including Tit-for-Tat opponent, which significantly differ from the trained opponents.

### D.3.2 *Monster-Hunt*

We use the policy population $\Pi_2$ trained by 4 $w$ values (i.e., $w = [5, 1, -5]$, $w = [4, 2, -2]$, $w = [0, 5, 0]$, $w = [5, 0, 5]$) in the RR phase as opponents for training the adaptive policy. In addition, we sample other 4 $w$ values (i.e., $w = [5, 0, 0]$, $w = [-5, 5, -5]$, $w = [-5, 0, 5]$, $w = [5, -5, 5]$) from $C_{max} = 5$ to train new opponents for evaluation. Fig. 20 shows the adaption training curve of the

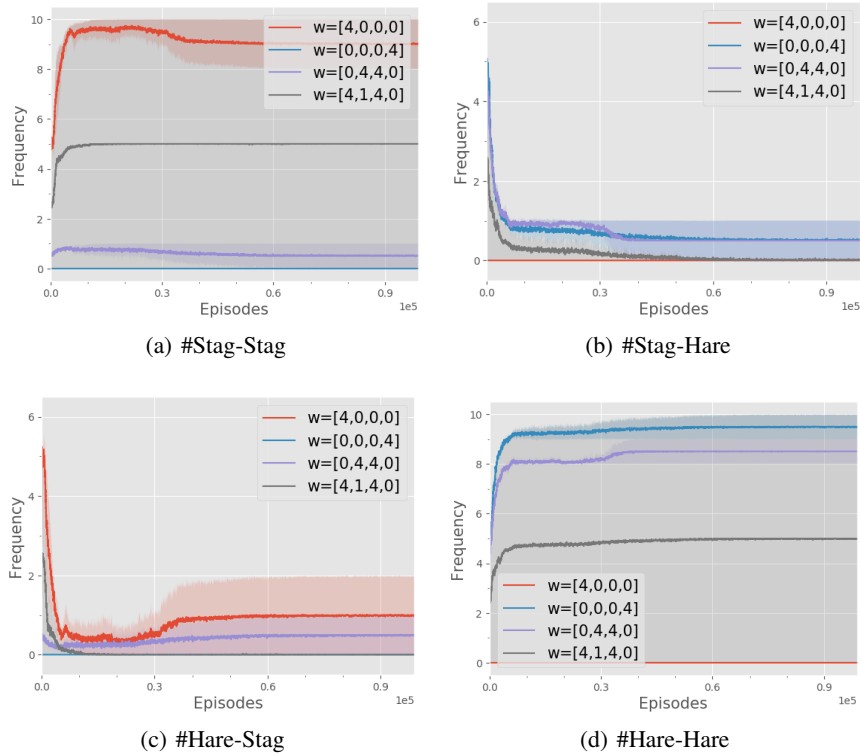

Figure 19: Adaption training curve in *Iterative Stag-Hunt*. #Stag-Stag: frequency of both agents hunting the stag. #Stag-Hare: frequency of agent1 hunting the stag while agent2 eating the hare. #Hare-Stag: frequency of agent1 eating the hare while agent2 hunting the stag. #Hare-Hare: frequency of both agents eating the hare. Frequency: times of certain behavior performed in one episode.

monster-hunt game, where the adaptive policy could take actions stably according to the opponent's identity.

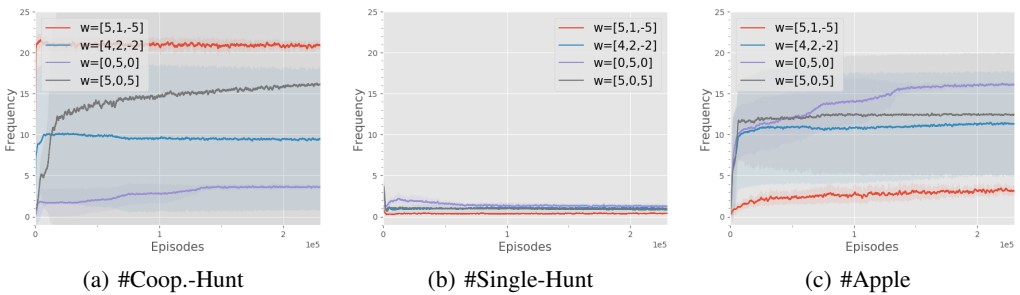

Figure 20: Adaption training statistics of *Monster-Hunt*. #Coop.-Hunt: frequency of both agents catching the monster; #Single-Hunt: the adaptive agent meets the monster alone; #Apple: apple frequency.

### D.3.3 *Agar.io*

In *Agar.io*, we used 2 types of policies from RR: $w = [1, 0]$ (i.e. cooperative) and $w = [0, 1]$ (i.e. competitive) as opponents, and trained a adaptive policy facing each opponent with probability=50% in *standard setting* while only its value head could know the opponent's type directly. Then we supposed the policy could cooperate or compete properly with corresponding opponent. As Fig. 21 illustrates, the adaptive policy learns to cooperate with cooperative partners while avoid being exploited by competitive partners and exploit both partners.

**More details about training and evaluating process:** Oracle pure-cooperative policies are learned against a competitive policy for 4e7 steps. So do oracle pure-competitive policies. And the adaptive policy is trained for 6e7 steps. the length of each episode is 350 steps (the half is 175 steps). When evaluating, The policy against the opponent was the adaptive policy in first 175 steps whatever we are testing adaptive or oracle policies. When we tested adaptive policies, the policy against the opponent would keep going for another 175 steps while the opponent would changed to another type and its hidden state would be emptied to zero. When we tested oracle policies, the policy against the opponent would turn to corresponding oracle policies and the opponent would also changed its type while their hidden states were both emptied.

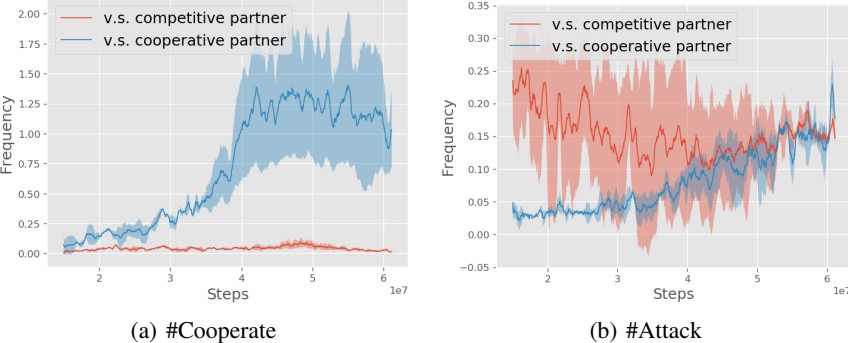

(a) #Cooperate          (b) #Attack

Figure 21: Statistics of adaptation experiments of *Agar.io*. (a),(b) illustrate frequencies of *Cooperate* and *Attack* when the adaptive policy was facing different partners. In (a), we can see that the agent learned to cooperate when the partner was cooperative; In (b), the descend of the "v.s. competitive partner" line at the beginning indicates that the adaptive policy was learning to avoid being exploit; The rising of both lines in the end indicates that the adaptive policy was also learning to exploit its partner.

