# OpenReview forum: "Discovering Diverse Multi-Agent Strategic Behavior via Reward Randomization"
_ICLR.cc/2021/Conference — ICLR 2021 Poster_

### Official Review · AnonReviewer3 · 2020-10-22
**The paper is interesting but it could be improved**

**Rating:** 6
**Confidence:** 4

**Review:**

This paper proposes to use reward randomization to explore the policy space in multi-agent games. The idea is that in most of multi-agent games multiple Nash Equilibriums exist with different payoffs. The goal is to find the NE that provides the highest payoff.
Policy Gradient and its variants, which have obtained a lot of practical successes, in general fail to find the NE with the highest payoff.
A first approach could be to re-start PG with different initializations for finding different NEs and then selects the best one.
In contrast, the authors propose to randomize the reward structure for exploring different policies. Then the policies are optimized on different reward structures with PG. The policy that leads to the highest payoff is selected and then optimized with PG on the original structure of rewards.
The authors provide some theoretical results to show that reward randomization has a highest probability to find the best NE than random initializations of PG.
The authors also propose to use reward randomization for learning an agent against different type of opponents.
The experiments are done on three games and show the interest of their approach in comparison with several baselines.

The paper is well written, proposes interesting ideas supported by analytical and experimental results. However the reviewer has some remarks, concerns and questions.

Concerning Theorem 1, O(\epsilon) for a probability is not a strong result: it can be higher than 1. After looking the proof, the reviewer thinks that it seems possible to provide the right expression of the probability of finding the high payoff NE.

Concerning Theorem 2, the proof is quite informal and the reviewer is not sure that it is correct. In particular, it is not clear if the same condition than in Theorem 1 is necessary: a-b = \epsilon (d-c). In the statement it seems not because a,b,c,d are uniformly sampled and there is no \epsilon in the statement, but the remark stating that RR necessitates at most O(\log1/\epsilon) times to achieves 1-\epsilon suggests that it does.
Moreover the reviewer thinks that the proposed analysis (statements of Theorem 1 and 2) will be more convincing if the number of starts, needed by the two approaches for finding w.h.p the high payoff NE, is compared (as you did in the remark).

In Algorithm 2, the authors write that the policy \pi’_2 is drawn from \Pi_2, but in the experiments section 5.3, the authors explain that \Pi_2 is carefully built, meaning that the policies in \Pi are chosen to be effective. This step is not in Algorithm 2, which is still correct, but this suggests that if \Pi_2 is not well chosen Algorithm 2 does not work.
This leads to my main concern. The rewards seem to be uniformly sampled with the constraint that their sum is no more than C_{max}. However, with this kind of uniform sampling the set of games used for exploring policies contains a lot of games that does not respect the constraints induced by the original game M. For instance in stag-hunt we have a \geq b \geq d > c. Using uniform sampling most of the induced games do not respect this reward structure. So it can lead to inefficient policy. For instance if a < b and a < d an efficient policy is to not track the stag and to hunt the hare. The reviewer understands that the diversity of rewards allows the diversity of obtained policies, but the reviewer is wondering if sampling the rewards with respect to the reward constraints of the game is not enough to obtain the diversity of policies. At a minimum, it could be interesting for the reader to have this reasonable baseline. By the way, may be this baseline allows Algorithm 2 working without carefully choosing the set of policies \Pi.

Overall, the paper is interesting, but the reviewer thinks that it could be better. The reviewer can change his mind if his concerns are answered.


___________________________


After the rebuttal I raised my score.

---

> ### Author Response · Authors · 2020-11-13
> **We have revised our paper accordingly.**
>
> Thanks for the valuable comments.
>
> Regarding “theorem 1”, thank you for the suggestion. We have updated the paper with the precise failure probability.
>
> Regarding “theorem 2”, We are sorry about the confusion in the previous version. We have updated the paper to make this point clearer. You are right that RR is independent of $\epsilon$. In the new version, we clarify that to achieve a $1 - \delta$ success probability where $\delta$ can be different from epsilon, RR needs at most $O(\log 1/\delta)$ whereas naive PG requires at least $\Omega ( 1/\epsilon * \log 1/\delta)$.
>
> Regarding “In Algo 2, $\Pi_2$ is carefully built”, we are sorry for such an inaccurate algorithm description in the previous version. Now the description in Sec. 3.1 has been corrected. We clarify that $\Pi_2$ is *NOT* "carefully built". In fact, $\Pi_2$ is simply a subset of the entire population. The reason is that the policies derived by RR typically converge to a few NEs. For example, in the standard Agar setting, The entire population simply converges to two NEs: a cooperative strategy mode and a non-cooperative mode. We take a subset of the population to make training faster. As we now state in the paper, if we make sure $\Pi_2$ covers all the modes in the RR population, even a subset of the population is sufficient for building adaptive agents.
>
> Regarding “the sampled reward function does not respect the original constraint”: at a high level, note that the original game of interest is typically a “challenging” case for MARL (otherwise we won’t study it at all), so sampling random stag hunt games instead of purely random games makes learning more difficult. As a specific example, considering the matrix-form stag-hunt game, if we only sample random stag-hunt games, it can be verified that the failure probability at each random restart can only be bounded by 0.66. This is worse than pure uniform sampling whose failure probability is smaller than 0.6. In addition, for complex Markov games, it can be non-trivial to even understand what the constraint is. For example, in the Agar game, we can qualitatively notice that there exists a risky cooperative strategy and a safe non-cooperative strategy but it requires some non-trivial computation to derive the exact constraint on the reward structure to ensure sure such a stag-hunt dynamic exists. Hence, in practice, we would suggest simply perform uniform sampling. Note that we only care about what final strategy we obtained w.r.t. the original game of interest. Breaking the constraint during the RR phase would only promote diversity without hurting the final empirical performances.

---

> > ### Comment · AnonReviewer3 · 2020-11-19
> > **a step forward**
> >
> > I thank the authors for their answers to my concerns.
> >
> > Concerning Theorem 1, it is ok.
> >
> > Concerning the proof of Theorem 2, I did not get where 0.6 comes from in equation 2. Could you detail?
> >
> > Concerning Algorithm 2, thank you for the precision.
> > I suggest you to give more details about the way where \Pi_2 is obtained from P.
> > Do you choose one policy per NE \in P? Do you choose only \pi_2 corresponding to a NE? ...
> > I think this step is crucial for Algorithm 2.
> > It has to be included inside Algorithm 2, and P should be the input rather than \Pi_2.

---

> > > ### Author Response · Authors · 2020-11-19
> > > **We have further updated our paper**
> > >
> > > Regarding theorem 2, note Equation (2) is an integral over 6 variables, so we can just use numerical integration software to evaluate it.
> > >
> > >
> > > Regarding Algo 2, thanks for the comments, we have further updated our paper (Sec. 3.1) to better clarify the details.
> > >
> > > Algo 2 aims to construct an adaptive policy by training with diverse opponents. So as $\Pi_2$ contains more diverse strategies, the agent generalizes better. Setting $\Pi_2=\mathcal{P}$ is definitely a natural choice. However, since $\mathcal{P}$ typically only contains a few strategic modes, it is *unnecessary* to include every ''duplicate'' policy from $\mathcal{P}$. Moreover, from an optimization perspective, a larger $\Pi_2$ makes optimization more challenging, so we can make learning easier by removing ''duplicate'' strategies. Hence, a practical solution for $\Pi_2$ is to simply include *at least* one policy from every discovered equilibrium from $\mathcal{P}$. To be more concrete, in the Monster-Hunt game (Sec 5.3, part 1), since we discovered 4 modes, we choose 1 policy from each mode to construct $\Pi_2$ (1 policy per mode) and the learned policy adapts well; in the standard Agar game (Sec 5.3, part 2), since we discovered 2 modes, we simply choose 2 cooperative policies and 2 competitive policies to construct $\Pi_2$ (2 policies per mode) and the trained agent achieves good performances as well.

---

### Official Review · AnonReviewer2 · 2020-10-27

**Rating:** 7
**Confidence:** 2

**Review:**

This work proposes and evaluates reward randomization as a strategy to discover efficient policies in multi-agent games. By perturbing the reward structure of a game, the authors show that policy gradient techniques can find better equilibria and lead to complex new behaviors.

Reward randomization is a fairly intuitive technique. In matrix form games, this just involves replacing the game's rewards with those drawn from some distribution. In more complex games, the authors propose distilling the game's reward structure into a small number of components and randomizing the weights placed on those components to compute the total payoff. The authors demonstrate that both theoretically and empirically, Reward-randomized Policy Gradient (RPG) outperforms standard baseline techniques. It also produces a diverse set of candidate policies, which the authors demonstrate can be used to train an adaptive agent.

The paper is clear and well-written. The figures are helpful and detailed. Overall, my impression is positive.

Some analysis of the sensitivity of this method would have helped. Much of the complexity is folded into choosing an appropriate reward distribution, and it would be good to know how dependent the results are on being able to choose a "good" distribution from which to sample.

I would have appreciated more of a discussion about why reward randomization works. Fundamentally, this appears to rely on the idea that while the action space may in general be very large, for most natural games, there are no reasonable reward schemes that incentivize the vast majority of these behaviors. As a result, it is feasible to explore the space of potentially optimal policies simply by exploring the space of reasonable rewards. Can this statement be formalized? And how well does it generalize to games other than the ones considered?

---

> ### Author Response · Authors · 2020-11-13
> **Thanks for your comments**
>
> We appreciate your valuable comments.
> Regarding the “sensitivity test” and “what is a ‘good’ distribution to sample rewards”, it is a great question! We use uniform sampling since it is the most basic and natural choice and even using such a simple sampling scheme, we have already achieved good empirical performances and interesting emergent behavior. Note that in the games we considered, the reward function has a pretty low-dimensional feature vector (at most 4), so random sampling is sufficient to fully explore the reward function space. We are investigating more complex multi-agent games and will tackle this question in our next project.
>
> Regarding “the statement to be formalized”, yes, your understanding is correct. We have updated our paper with more discussions and remarks. Intuitively, there are two fundamental reasons why reward randomization works:
> (1) the first perspective is as you stated, the possible action sequence is exponentially large while the reward structure of a game is typically low-dimensional. Also, the number of possible NEs in a game is often small;
> (2) the second is from an optimization perspective. The intuition has been already illustrated in Fig 1. Note that PG is a local improvement algorithm. This implies that the policy needs to first reach the neighborhood of an NE in order to converge to it via MARL. However, due to the reward structure, in many challenging games, the neighborhood of the optimal NE is extremely small (e.g., trust dilemmas), which poses substantially hard exploration challenges for the MARL algorithm. However, if we can perturb the reward structure properly (e.g., sample from a proper reward function space), PG may easily converge to a policy that belongs to the optimal NE neighborhood in the original game. Subsequently, by evaluating and fine-tuning the derived policy in the original game, we can obtain the optimal NE easily since the action-space exploration is now executed in a much easier reward landscape.
>
> Regarding “generalize to other domains”, we have tested 3 challenging trust dilemmas including a real-world game, Agar.IO. Our algorithm is not domain-specific. It can be potentially applied to any domain with a proper reward structure. We will leave more extensions to a wider range of real-world games as future work.

---

### Official Review · AnonReviewer4 · 2020-10-28
**Successful application of reward perturbation techniques to identify high-value equilibria that offers limited insights**

**Rating:** 5
**Confidence:** 3

**Review:**

The paper focuses on an important problem, the existence of multiple Nash equilibria in strategic games, and proposes a relatively simple mechanism, reward randomization, which allows agents to discover higher-payoff equilibria in games where normal decentralized policy gradient methods would converge to suboptimal ones. What is proposed is to perturb actual rewards with random perturbations, which essentially, to my mind simply creates a range of games agents find themselves in so that the chances are increased that policy search will end up finding more efficient equilibria.

While the overall idea is interesting, the paper actually lacks a precise problem definition that would also link it to the (huge) literature in the area, which it does refer to, but actually not directly compare to, as the now common deep learning route to solving the fundamental (and notoriously hard) problem is applied, which operates in a completely different setting - training function approximators using an enormous number of games played offline. This cannot be compared to the problems considered in game theory and traditional multiagent reinforcement learning, where policies are learned online from a relatively small number of examples, and an algorithm needs to be able to perform well against very broad classes of opponents, without the benefit of training itself on modified problems pre-play.

That said, I appreciate there is value to the theoretical results that provide some more general evidence for the potential importance of the method.

The paper makes a lot of assumptions about structure in larger games in order to apply feature-based learning approaches, but one could have approached the whole problem with a simple game and then simply demonstrate how it scales up - it seems like there is not that much to say about the conceptual ideas when one reads all the details.

An excessive part of the paper is spent on explaining ideas in the stag hunt game and describing the other games, with a lot of further detail included in the lengthy supplementary material, but much of this seems to detail the very specific process of applying the technique and competing alternatives in a few specific games - I don't think much of this adds to our understanding of the problem.

Beyond these criticisms, the paper is generally clearly written, and appears technically sound.

---

> ### Author Response · Authors · 2020-11-13
> **Paper has been revised to clarify our problem definition, focus, and assumption**
>
> Thanks for your valuable comments. We have updated our paper accordingly.
>
> Regarding the “problem definition”, we have substantially expanded Sec. 3 to more precisely explain the problem definition.
>
> We also want to clarify that we do have a *different focus* from the existing literature of game theory. Understanding general-sum games is extremely challenging and remains a long-standing problem in multiple areas. We primarily focus on how to improve the *empirical* performance of MARL algorithms, particularly on those failure cases we have noticed and studied in this paper (that’s why we didn’t put game theory or NE in keywords). We also hope our analysis and insights can at least bring new directions to the MARL community for further research ---- there are very few works that have considered games similar to what we presented. Even the problem itself has not yet attracted sufficient attention from the MARL community although it is important. We hope our paper can bring new perspectives and insights into the MARL community to be aware of such a broad range of problems from game theory.
>
> Our paper presentation follows more the convention of MARL works, which typically include a large portion of experiments and implementation details. Though our method is inspired by the analysis of the stag hunt game, our primary focus is to provide an empirical solution to tackle Markov games. Hence, we experiment on 3 complex temporal trust dilemmas and show interesting emergent behaviors. Traditional game theory tools typically have difficulties to analyze these complex temporal games, particularly the real-world game Agar.IO. We also assume problem simplifications, e.g., due to the strong symmetric payoff structure in temporal trust dilemmas, we don’t need to tackle the NE refinement problem while we do respect the literature in Sec. 6.
>
> Lastly, we also want to emphasize that we introduced a new multi-agent environment, Agar.IO, for the community. Our simulator strictly follows the original popular Agar.IO game. Although our paper only considers a 2-player setting, the simulator is designed for massive players, which allows complex multi-agent behaviors and could benefit a wider community for future research.

---

### Official Review · AnonReviewer1 · 2020-10-29
**Important problem, but proposed algorithm probably makes implicit assumptions?**

**Rating:** 6
**Confidence:** 3

**Review:**

This paper considers the problem of finding a nash equilibrium in two player games where each of the algorithm runs an RL algorithm. In this paper they ask the question -- which nash equilibria does the dynamics converge to in this two player game (where each player optimizes based on a policy gradient algorithm). They construct two player games with multiple nash equilibria; one is a favorable nash equilibria where both players get high rewards while the other is a less favorable nash equilibria where both player only get medium rewards. In such games they first show that in general simply running policy gradient on the natural reward function i.e., the observed payoff will not lead to the desirable nash equilibria. The goal of this paper is to ameliorate this by considering perturbations in the reward space. At a high level, the algorithm learns multiple policies on a class of games generated by sampling multiple reward functions from a family and training one policy per sampled reward function using PG. Then using an evaluation function, the best policy is picked by evaluating each of the learnt policies on the original game.

My comments on this paper are as follows. First, I think the question studied in this paper is well-motivated. In general, controlling for which nash equilibrium a two player (or N-player in general) dynamics should converge to is an important problem. This problem has been extensively considered in the game theory and online learning literature. So the importance naturally extends to the multi-agent RL world. The proposed algorithm here is reminiscent of follow-the-perturbed-leader, where perturbations in the reward space (as opposed to the policy space) leads to improved algorithms. The strengths of this paper are as follows.

- The initial parts of the paper are well-written. They consider simple toy examples to show that PG can indeed lead to bad equilibria. Then using some stylized analysis, show that sampling from the reward space can indeed overcome this problem (in games where the family of reward functions for which PG converges to the desired nash equilibria is large, yet for the specific game at hand PG leads to a bad equilibria).
- Extensive experimental evaluation. The paper considers a total of four test bed games and shows the benefit of using reward randomization in these settings.  In the supplementary materials there is extensive simulated results on these games. In particular, they fix a particular "true" payoff matrix and then run their algorithm against a few baselines assuming this game.
- The problem considered is important, not just in the artificial game setting but also in many practical applications which can be modeled as a game that ends up having multiple NE.
- Specification of the hyper-parameters of the algorithm used. Also evaluating the algorithms using multiple seeds (which is an important criteria for RL algorithms).

Having said that, here are some of the weakness of this paper.

- The considered algorithm "works" probably because the settings are easy (As defined by convergence to the correct NE by PG algorithms) for an average instance from the space of all instances. In particular, consider the following thought exercise. Let us sample a random payoff matrix from the space of payoffs and run PG on this game. What is the prob. that PG will get stuck at the bad NE? If the answer is that this prob. is high, then it is also likely that the proposed algorithm will not work (since this is essentially the idea being exploited in the algorithm). On the other hand, if the prob. is low then it shows that in general the considered game is easy and this algorithm only optimizes for the "outlier" scenarios. In general, that is the biggest weakness I see in this paper; it makes a pretty strong implicit structural assumption on the games and their corresponding payoff landscape. Am I missing something here?
- I also think this paper could improve its presentation. First, the experimental setup is pretty unclear. What is the reward function in each of the games. I do not see a formal definition. Likewise, the algorithm description itself is pretty informal. It takes a bit of leap of faith in assuming that there exists a well-defined evaluation function (that is computable). It would greatly help the paper in readability if things are written in a formal manner. Moreover, for each of the games (possibly in supplementary) taking a detailed approach to the setup of the game, the reward function and the evaluation function would help in reproducibility and understanding of this paper. Finally, I also think some of the graphs could use better color-schemes or other differentiating factors apart from color. Some of the colors are pretty close to each other and in general harder to read.

My overall evaluation is based on the first point in my weakness section and considering the paper in totality. If the authors can sufficiently answer that question and show convincing experiments and/or explanation I am willing to change my score.

---

> ### Author Response · Authors · 2020-11-13
> **We have revised our paper to clarify the confusion**
>
> Thanks for your valuable comments. We have put more discussions and insights into the updated paper for easier understanding.
>
> Regarding the “thought experiment”, there are two critical factors that you probably miss:
> (1) The first critical point is the *definition* of a "good"/"bad" equilibrium. The "goodness" is defined w.r.t. a specific game. An NE that MARL gets stuck at a random game can be “bad” w.r.t. that random game. However, that particular NE can be possibly “good” in a game we care about (e.g. Stag Hunt). That being said, a bad policy in some game might be an extremely good one in another game with a different reward function. It doesn't really matter whether the MARL algorithm gets stuck at a “bad” NE or not in the average case since we only care about the solution we eventually obtained for the original game of interest. Our method can be viewed as an exploration technique that explores in the reward-space to discover novel strategies, which is conceptually similar to those action-space exploration methods but substantially more efficient.
> (2) Most real-world games we care about are not random games ---- real-world games have structures and it is often the case that we just care about those most challenging games, where MARL can hardly discover a “good” equilibrium (if MARL can easily solve that game, we won’t be interested in it). Our insight is that a hard-to-find equilibrium could be easy to discover in random games ------ even though those random games does not make sense to humans, the policy does. It is possible that our algorithm may not necessarily work for every (random) game but it does help in several extremely challenging games, i.e., stag-hunt, temporal trust dilemmas, etc. --- these are not outliers, instead, they are games that many people care about.
>
> Regarding the “reward function definition”, we apologize for the confusion. You can find the formal definition of the reward function, i.e., the associated feature vector and the weight, in the experiment section (Sec. 5). We have clarified this at the beginning of Sec. 4. The goal of Sec.4 is to highlight why those testbeds are challenging for MARL. We were hoping the visualization (Fig. 3,4,5) could have clearly illustrated the game rewards. We apologize for the confusion.
>
> Regarding “the evaluation function $E$”, $E$ is simply a *payoff* evaluation function, which can be any linear combination between the utility of two agents in our paper. We have clarified it in Sec.3 in the updated paper. Also, as stated in Sec 5, we simply use $E(\pi_1, \pi_2) = U_1(\pi_1, \pi_2)$ (the utility of agent 1) in our experiments since all the discovered (approximate) NEs have symmetric payoffs in this paper.

---

### Author Response · Authors · 2020-11-13
**Paper updated with changes in red; one contribution largely ignored.**

We have updated our paper to incorporate all the feedback from reviewers. All the changes are colored red.

The major changes are as follow:
1. We clarify our contributions in Sec.1:
        (1) we studied a collection of challenging multi-agent games;
        (2) a new reward-space exploration technique, reward randomization;
        (3) empirically show RPG can discover diverse strategies;
        (4) we released a new MARL environment, Agar.IO, which can benefit a wide research community.
    **We note that contribution (4) is largely ignored in all the reviews.**
2. We made theorems and proofs more rigorous and included more discussions.
3. We added more rigorous definitions of the problem we want to solve for general Markov games in Sec 3.
4. Fixed an inaccurate detail in Sec. 3.1

---

### Decision · Program_Chairs · 2021-01-07
**Final Decision**

**Decision:**

Accept (Poster)

**Comment:**

All the reviewers are in favor of accepting this paper, which demonstrates both theoretically and empirically the value of reward randomization in solving multi-agent reinforcement learning problems. The rebuttal phase was crucial in improving the quality and evaluation of the submission. I am glad to recommend acceptance.